# BRAIN-SEMANTOKS:
# LEARNING SEMANTIC TOKENS OF BRAIN DYNAMICS WITH A SELF-DISTILLED FOUNDATION MODEL

**Sam Gijsen**[1,2,3*]  **Marc-Andre Schulz**[1,2,3]   **Kerstin Ritter**[1,2,3]

[1]Hertie Institute for AI in Brain Health, University of Tübingen, Germany
[2]Tübingen AI Center, University of Tübingen, Tübingen, Germany
[3]Charité – Universitätsmedizin Berlin, Department of Psychiatry and Psychotherapy, Berlin, Germany
sam.gijsen@charite.de, marc-andre.schulz@charite.de, kerstin.ritter@uni-tuebingen.de

## ABSTRACT

The development of foundation models for functional magnetic resonance imaging (fMRI) time series holds significant promise for predicting phenotypes related to disease and cognition. Current models, however, are often trained using a mask-and-reconstruct objective on small brain regions. This focus on low-level information leads to representations that are sensitive to noise and temporal fluctuations, necessitating extensive fine-tuning for downstream tasks. We introduce Brain-Semantoks, a self-supervised framework designed specifically to learn abstract representations of brain dynamics. Its architecture is built on two core innovations: a semantic tokenizer that aggregates noisy regional signals into robust tokens representing functional networks, and a self-distillation objective that enforces representational stability across time. We show that this objective is stabilized through a novel training curriculum, ensuring the model robustly learns meaningful features from low signal-to-noise time series. We demonstrate that learned representations enable strong performance on a variety of downstream tasks even when only using a linear probe. Furthermore, we provide comprehensive scaling analyses indicating more unlabeled data reliably results in out-of-distribution performance gains without domain adaptation.

## 1 INTRODUCTION

The investigation of brain dynamics has been a cornerstone of neuroscience, progressing our understanding of human cognition, disease, and aging. Functional magnetic resonance imaging (fMRI) has been an instrumental modality in this endeavor; its blood-oxygen-level-dependent (BOLD) measurement relates to local changes in brain activity, and its non-invasive nature has made it a primary tool across numerous research fields (Ogawa et al., 1990; Logothetis, 2008). Despite being extremely high-dimensional, fMRI data is often collected in limited samples, which can severely constrain the potential insights (Button et al., 2013; Poldrack et al., 2017). This challenge motivates a shift towards data-driven representation learning, where progress in self-supervised learning (SSL) can enable the training of highly capable 'foundation' models from large quantities of unlabeled data, promising a new way forward for neuroimaging analysis.

Current fMRI foundation models, however, adapt reconstruction-centric paradigms from NLP and vision, such as masked signal prediction (Caro et al., 2023; Wang et al., 2025). While latent-space reconstruction (e.g., JEPA; Dong et al. (2024)) avoids modeling the substantial noise in the BOLD signal, these models still focus on low-level, regional information. We argue this objective is misaligned with predicting stable, high-level phenotypes, as the resulting representations require extensive fine-tuning for such tasks, reducing the utility of a foundation model. This dependency is particularly problematic for fMRI, where transfer is challenged by significant variations across datasets in participant cohorts, hardware, and acquisition protocols.

---

*We make code and models available at https://github.com/SamGijsen/Brain-Semantoks

To address these issues, we hypothesize that effectively predicting stable phenotypes requires a shift from reconstruction to abstraction. The goal should not be to perfectly encode the BOLD signal, but to abstract away from it to find the underlying phenotypic signature. We propose Brain-Semantoks, a foundation model built on a strong neuroscientific inductive bias to learn such abstract representations. Our approach starts at the input level, recognizing that self-attention mechanisms, central to modern transformers, perform best on sequences of low-noise, semantic tokens, akin to words in natural language. Time series of individual, small regions are poor tokens in this regard as they are noisy and lack high-level meaning. We therefore introduce a semantic tokenizer, which aggregates information from regions within a functional brain network (e.g., default mode network) into a single, robust token. This creates a shorter, more computationally efficient, and semantically meaningful sequence for the transformer to operate on.

With these semantic tokens, we then shift the learning objective itself. Instead of focusing on the reconstruction of masked signals, Brain-Semantoks is trained using a self-distillation objective to produce a stable, summary representation across different temporal views of the same scan (Grill et al., 2020; Caron et al., 2021). This explicitly trains the model to capture a stable, high-level representation of an individual's brain dynamics, which we expect to transfer better across data distributions. However, while conceptually highly suitable, we found that applying this objective to low signal-to-noise fMRI data can lead to training instability, where the model converges on a poor, simple solution. To solve this, we introduce a Teacher-guided Temporal Regularizer (TTR), a novel training curriculum active only at the start of training. This regularizer guides the model to first learn the time-averaged signature of each network before modeling more complex temporal variations, ensuring robust and meaningful pretraining convergence. The resulting representations are particularly powerful for linear probing, indicating they are well-disentangled and broadly useful without task-specific fine-tuning.

**Contributions.** Our contributions are threefold. First, we propose a new pre-training approach that prioritizes abstract representations over signal reconstruction, enabled by a novel semantic tokenizer and a Teacher-guided Temporal Regularizer (TTR) to stabilize training. Second, we introduce Brain-Semantoks, a foundation model trained with this method that achieves state-of-the-art performance on diverse downstream tasks under a rigorous linear probing protocol. Finally, we provide the first detailed scaling analysis for fMRI foundation models, showing consistent out-of-distribution performance gains without domain adaptation.

## 2 RELATED WORK

**Self-Supervised Learning for MRI.** Much early self-supervised learning work focused on reconstruction using auto-encoders (Han et al., 2019; Pinaya et al., 2019; Kim et al., 2021), using relatively limited data. The first effort to build an fMRI foundation model similarly adapted reconstruction-based objectives popular in other domains. BrainLM (Caro et al., 2023) employs a masked modeling objective to reconstruct the BOLD signal in input space. While effective, this approach risks modeling the substantial noise inherent in fMRI data. More recent work like Brain-JEPA (Dong et al., 2024) mitigates this by predicting masked representations in a latent space, thereby learning to ignore noise. Meanwhile, NeuroSTORM operates on 4D voxel data, performing spatio-temporal reconstruction (Wang et al., 2025). However, all these methods remain fundamentally focused on predicting low-level information. In contrast, models like BrainMass learn from static functional connectivity matrices (Yang et al., 2024), ignoring the rich temporal dynamics central to our work.

**Self-Distillation Learning.** Self-distillation has proven highly effective for learning semantic features, improving upon contrastive learning methods (Chen et al., 2020). Seminal works include MoCo (He et al., 2020), BYOL (Grill et al., 2020), and DINO (Caron et al., 2021; Oquab et al., 2023; Siméoni et al., 2025) demonstrated that a student network can learn powerful, linearly separable representations by matching the output of a teacher network (a momentum-updated version of itself) across different views of a sample. This approach avoids "representational collapse" without requiring negative samples or a reconstruction loss. The iBOT framework (Zhou et al., 2021) further advanced this by integrating a masked-token prediction objective within the distillation framework, enabling the model to learn both a global summary representation as well as rich, context-aware local features. Recent work by Wu et al. (2025) makes important progress in understanding what

is necessary to prevent collapse and significantly simplifying the approach, reducing the number of hyperparameters.

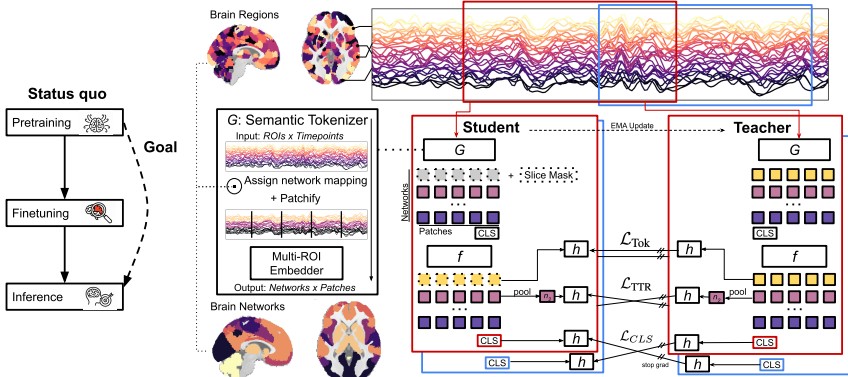

Figure 1: **Brain-Semantoks.** A student-teacher architecture is used to learn stable brain dynamics representations across time by aligning long temporal views. The semantic tokenizer ($G$) is used to produce robust tokens of functional brain networks, which serve as input to a transformer encoder ($f$). Three losses are used during pretraining: a temporary regularisation loss for stability ($\mathcal{L}_{TTR}$), a within-view, latent space prediction loss of masked tokens ($\mathcal{L}_{Tok}$), and a global cross-view loss to learn a high-level, semantic representation ($\mathcal{L}_{CLS}$).

# 3 METHOD: BRAIN-SEMANTOKS

Our proposed framework, Brain-Semantoks, learns abstract and temporally stable representations from fMRI time series. The methodology is built upon three core innovations designed to address the unique challenges of fMRI data. First, we introduce a paradigm performing self-distillation across time, that explicitly trains for high-level representations suitable for transfer learning. Second, we develop a semantic tokenizer with a strong neuroscientific inductive bias to create a robust and meaningful input space for our encoder model. Finally, we introduce a training curriculum that stabilizes the learning objective, ensuring convergence on low signal-to-noise data. The framework uses a student-teacher architecture, as depicted in Figure 1.

## 3.1 A SELF-DISTILLATION FRAMEWORK FOR SEMANTIC REPRESENTATIONS

The primary goal of a foundation model is to learn representations that are broadly applicable without requiring task-specific fine-tuning. To achieve this with fMRI, a model must learn to capture the stable, underlying phenotypic signature of a subject, abstracting away from transient noise and acquisition-specific artifacts.

**Input Data and Augmentation:** We represent a subject's fMRI time series as matrix $X \in \mathbb{R}^{C \times T}$, where $C$ is the number of brain regions of interest (ROIs) and $T$ is the number of time points. To generate different views of the same underlying brain dynamics we create two long temporal segments of length $T_{crop} < T$ resulting in two views, $X^{(1)}$ and $X^{(2)}$. Unlike computer vision, a large set of intuitive augmentations are not available for fMRI. We therefore mainly rely on self-distillation across time and only lightly further augment the views with corrupting transformations: we randomly select a fraction of channels ($\tau_c$) and contiguous timepoints ($\tau_t$) and set them to zero, add gaussian noise sampled with $\mu = 0$ and $\sigma = \tau_\sigma$, and finally scale the time series amplitude $X\tau_s$.

**Student-Teacher Framework:** We use a student-teacher architecture to enforce representational consistency across these two views. The student network $f_s(\theta_s)$, is trained to match the output of the teacher network $f_t(\theta_t)$. The teacher provides a stable regression target as its weights are an exponential moving average (EMA) of the student's weights:

$$\theta_t \leftarrow \alpha\theta_t + (1 - \alpha)\theta_s \tag{1}$$

where the momentum coefficient $\alpha$ gradually increases during training. This forces the student to learn high-level representations which are stable across time.

## 3.2 SEMANTIC TOKENIZER

We posit that standard tokenization such as a direct linear projection of ROI signals is suboptimal for fMRI data. This approach creates overly long sequences of noisy, low-level tokens that hinder a transformer's ability to learn meaningful long-range dependencies. Our innovation is a semantic tokenizer, $G(\Phi)$, that addresses this by creating a compact and robust input space grounded in a core neuroscientific prior: the brain's organization into functional networks. For a given fMRI scan $\mathbf{X}^{(v)}$, our tokenizer uses $N$ independent, network-specific modules, $g_n$, each operating on the time series of a single network. Specifically, each $g_n$ processes the subset of ROIs belonging to network $n$, denoted $\mathbf{X}_n^{(v)} \in \mathbb{R}^{C_n \times T}$, to produce a sequence of $P$ semantically rich, $D$-dimensional tokens, $\mathbf{Z}_n^{(v)}$.

To model the BOLD signal's complex temporal structure, each module $g_n$ first divides its input time series into $P$ relatively long temporal patches $\mathbf{x}_p \in \mathbb{R}^{C_n \times (T/P)}$, inspired by the temporal stability of macro-scale brain dynamics (Allen et al., 2014; Vidaurre et al., 2017). Within each patch, a multi-scale convolutional filter bank, composed of a standard convolutional branch, $\text{Conv}_{\text{std}}(\cdot)$, and a structured convolutional branch, $\text{Conv}_{\text{str}}(\cdot)$ (Li et al., 2022), captures hierarchical temporal patterns. For more details on the implementation of these branches, please see Appendix A.3.

A final $D$-dimensional token embedding for each patch is generated via the transformation: token $=$ LayerNorm $(\text{AvgPool} (\text{GELU} ([\text{Conv}_{\text{std}}(x_p); \text{Conv}_{\text{str}}(x_p)])))$. The complete token tensor is then formed by concatenating the outputs from all network modules, $\mathbf{Z}^{(v)} = [\mathbf{Z}_1^{(v)}; \ldots; \mathbf{Z}_N^{(v)}]$, resulting in a final tensor of shape $\mathbb{R}^{N \times P \times D}$. This approach yields a compact and semantically rich input that provides the transformer encoder with a better starting point for learning, which is crucial for the stability of the self-distillation framework.

## 3.3 TRANSFORMER ENCODER AND MASKING

The sequence of network-tokens $\mathbf{Z}^{(v)}$ is flattened into a sequence of length $N \times P$. We add sinusoidal positional embeddings to encode the temporal order of the patches and a learnable network-specific embedding for each of the $N$ networks to encode their identity. Finally, a learnable [CLS] token is prepended to the sequence, yielding a sequence of length $N \times P + 1$.

The student and teacher models each consist of two core components: a transformer encoder backbone, $f(\theta)$, and a projection head, $h(\psi)$. The backbone, $f$, processes the input token sequence to produce network and [CLS] embeddings. These embeddings are then processed by the projector, $h$, into the space where the distillation loss is minimized. The projector is used only during pre-training and is discarded thereafter, while the transferable representations are the outputs from the teacher's encoder backbone, $f_t(\theta_t)$.

The student network, $f_s(\theta_s)$, receives a masked version of the token sequence. Masking is determined by a binary mask $\mathbf{B}^{(v)} \in \{0, 1\}^{N \times P}$, which replaces tokens with a learnable mask embedding. The teacher network always receives the full, unmasked sequence. To reduce the degree to which the model can rely on simple, interpolative relationships to predict masked tokens, we perform 'slice masking'. Specifically, we treat the input network-tokens as a 2D matrix of size $N \times P$ (networks by temporal patches) and mask out entire 'slices' rather than random individual tokens. We randomly select one of two strategies on a sample-level to increase data diversity. The first, network slicing, masks entire rows, hiding all temporal data for one or more selected networks. Second, temporal slicing masks a contiguous block of entire columns, hiding information from all networks for a specific period. By masking large, contiguous parts of data, we force the model to learn more complex relationships between networks and across time.

## 3.4 PRETRAINING OBJECTIVE

Finally, we propose a multi-component objective function that includes a curriculum to ensure stable training. All loss components are computed using the outputs of the projection heads ($h_s$ and $h_t$), which are denoted as $\mathbf{z_s}$ and $\mathbf{z_t}$. Following recent work simplifying DINO (Wu et al., 2025), we regularize with a coding rate term to prevent representation collapse.

To learn a high-level, stable representation of the brain time series, we enforce consistency between the [CLS] tokens across two views. The loss is bidirectional and regularized:

$$\mathcal{L}_{CLS} = \mathbb{E}\left[d(z_{s,\text{CLS}}^{(1)}, z_{t,\text{CLS}}^{(2)}) + d(z_{s,\text{CLS}}^{(2)}, z_{t,\text{CLS}}^{(1)})\right] - \gamma R_\epsilon(\text{Cov}[z_{s,\text{CLS}}]) \tag{2}$$

where $d$ is the squared Euclidean distance and $\mathbb{E}$ is the expectation over data. Adopting the SimDINO implementation (Wu et al., 2025), we use the coding rate regularizer $R_\epsilon(\Sigma) = \frac{1}{2}\log\det(I + \frac{D_{\text{proj}}}{\epsilon}\Sigma)$ with $\epsilon = 0.05$ and where $D_{\text{proj}}$ is the projected feature dimension. This prevents subspace collapse by maximizing the determinant of the batch-estimated covariance $\Sigma$. To normalize regularization strength across varying batch sizes $B$, we adopt the balancing heuristic $\gamma = \frac{D_{\text{proj}}+B}{D_{\text{proj}}B}$.

**Network Token Loss** To promote the model learning rich, temporally-sensitive representations, we apply an auxiliary distillation loss on the network tokens that were masked in the student's input, guided by the 2D mask matrix $\mathbf{B}^{(v)}$ (Zhou et al., 2021). This loss is computed within each view:

$$\mathcal{L}_{\text{Tok}} = \mathbb{E}_{v\in\{1,2\}}\left[\sum_{n=1}^{N}\sum_{p=1}^{P} B_{n,p}^{(v)} \cdot d(\mathbf{z}_{s,n,p}^{(v)}, \mathbf{z}_{t,n,p}^{(v)})\right] \tag{3}$$

Due to the semantic tokenizer, this task is performed on a more semantic network-level, rather than on a noisier, lower region-level.

**Teacher-guided Temporal Regularizer**

Although our tokenization strategy significantly improves pretraining effectiveness, we found that a direct application of these distillation objectives can still lead to poor solutions with noisy fMRI time series. We therefore develop a principled stabilizing curriculum based on this observation.

Specifically, as we observe that a more compact token sequence aided convergence and yielded representations with better predictive performance, we guide the student network to first learn the time-averaged representation of each network. Conceptually, this constrains the token space $N \times P + 1$ towards $N + 1$, which helps find a good initial representation which can thereafter be refined with temporal variability. This can be directly adopted in the distillation framework by using the teacher model to provide the network-specific targets. The summary token for each network $n$ is computed by averaging its $P$ patch embeddings from the transformer output:

$$\bar{\mathbf{z}}_{t,n}^{(v)} = \frac{1}{P}\sum_{p=1}^{P}\mathbf{z}_{t,n,p}^{(v)} \tag{4}$$

The regularisation loss is then applied across views to these $N$ summary tokens:

$$\mathcal{L}_{\text{TTR}} = \mathbb{E}\left[\sum_{n=1}^{N}\left(d(\bar{\mathbf{z}}_{s,n}^{(1)}, \bar{\mathbf{z}}_{t,n}^{(2)}) + d(\bar{\mathbf{z}}_{s,n}^{(2)}, \bar{\mathbf{z}}_{t,n}^{(1)})\right)\right] - \gamma \sum_{n=1}^{N} R_\epsilon(\text{Cov}[\bar{\mathbf{z}}_{s,n}]) \tag{5}$$

The contribution of this regularizer is decayed to zero early in training to stabilize initial training without overly constraining the final learned solution.

**Total Loss Function**

The final training objective is a weighted sum of the three hierarchical loss components:

$$\mathcal{L}_{Total} = \mathcal{L}_{CLS} + \lambda_{Tok}\mathcal{L}_{Tok} + \lambda_{TTR}\mathcal{L}_{TTR} \tag{6}$$

where $\lambda_{Tok}$ and $\lambda_{TTR}$ are scalar hyperparameters balancing the contributions of different levels of self-supervision. Following training, the student weights are discarded while the teacher weights are used for downstream evaluation.

## 4 EXPERIMENTAL SETUP

### 4.1 DATASETS

**Pretraining Data**

We leveraged the largest 3T resting-state fMRI corpus available for unlabeled pretraining. Specifically, we use 39139 preprocessed recordings from the UKBioBank as well as the participant age and sex variables (UKB; Miller et al. (2016); application number 25163. We held out 1625 recordings for downstream evaluation.

We extract parcel-wise time series using the cortical Schaefer-400 atlas (Schaefer et al., 2018), subcortical Tian-III atlas (Tian et al., 2020), and cerebellar Buckner-7 atlas (Buckner et al., 2011), yielding 457 total ROIs. We follow fMRI preprocessing conventions and apply a 0.01-0.1Hz bandpass filter. Data normalization is a crucial step to aid transfer learning. Whereas prior work has relied on robust scaling (Caro et al., 2023; Dong et al., 2024), which preserves ROI-specific 'DC' offsets resulting from the fMRI scanner, it impairs transferability to datasets which have less or no such offsets. We therefore applied z-scoring to each ROI per scan. As the UKB's temporal resolution (0.735s) is higher than most available datasets, we temporally downsample to 2s, resulting in 180 timepoints for the 6 minute recordings. Both normalization and resampling happen on the parcellated time series, meaning these are light operations that can be performed online during data loading and thereby ease transferability.

**Downstream Tasks**

We construct a varied set of prediction tasks with differing sample sizes across multiple datasets (Appendix Table 9). Continuous targets are binned into multi-class labels to enable direct comparison between linear probing and finetuning.

For internal evaluation, we use UKB data for sex prediction and five-class age prediction (bins with equal sample sizes). For external evaluation: SRPBS (Tanaka et al., 2021) enables binary classification of schizophrenia and major depressive disorder versus controls; ABIDE (Craddock et al., 2013; Di Martino et al., 2014) provides autism-spectrum disorder classification; HBN (Alexander et al., 2017) allows prediction of language (CELF) and cognitive (WISC) scores using three bins, plus out-of-distribution demographic prediction; LEMON (Babayan et al., 2019) provides mood (MDBF) and cognitive task scores (CVLT, TMT) with three bins each. For scaling analyses, we additionally use demographic data from SRPBS and ADHD200 (Bellec et al., 2017). See Appendix A.2 for more detailed scale descriptions.

In total, we evaluate on four demographic, three clinical diagnosis, three cognitive/language, and one mood prediction task. Critically, these datasets vary in participant cohorts, acquisition protocols, and preprocessing. Time series standardization consists of resampling to 2s, band-pass filtering, and z-scoring.

## 4.2 IMPLEMENTATION

**Training and Model Architecture**

We sample temporal views of length $T_{crop} = 100$ timepoints. At a sample level, we apply light augmentations by randomly zeroing out a fraction of channels $\tau_c \sim \mathcal{U}[0, 0.1]$ and contiguous timepoints $\tau_t \sim \mathcal{U}[0, 0.3]$, adding Gaussian noise with $\sigma = \tau_\sigma = 0.1$, and scaling the amplitude by $\tau_s \sim \mathcal{U}[0.8, 1.2]$. The semantic tokenizer $G$ maps an input time series $\mathbf{X} \in \mathbb{R}^{C \times T_{crop}}$ to a token tensor $\mathbf{Z} \in \mathbb{R}^{N \times P \times D_f}$ with a patch length of 20. We define $N = 9$ functional networks, based on the Yeo 7-network parcellation for the cortex (Yeo et al., 2011), with two additional networks comprising all subcortical and cerebellar ROIs, respectively. Within each network-specific tokenizer $g_n$, the standard and structured convolutional branches each output features of dimension $D_f/2$, which are then concatenated.

The transformer encoder $f$ uses a dimensionality $D_f = 768$ with 8 layers. The projection head $h$ is an MLP with 2 hidden layers ($D_h = 1024$) and an output layer projecting down to $D_{\text{proj}} = 128$ dimensions and applying $\ell_2$-normalization. The head is shared across all three distillation objectives. We find it suffices to set $\lambda_{TTR} = 0.5$ (i.e., weighting of $\mathcal{L}_{TTR}$) and cosine-decay this weighting to zero over the first 5% of training steps. We use slice-masking with $\lambda_{Tok} = 0.5$ (weighting-factor) and a high masking ratio sampled from $\mathcal{U}[0.65, 0.85]$. For all analyses we use the model checkpoint following 100 epochs of pretraining. We provide further optimization hyperparameters in Appendix Table 17. We jointly train the tokenizer and encoder end-to-end. The reduced memory usage of the tokenizer enables pretraining in under two hours on a single GPU with less than 20 GB of memory.

Table 1: Balanced accuracy (%) for fMRI time series foundation models using linear probes. Mean $\pm$ standard deviations. Best results in **bold**, second-best underlined. †Significantly worse than best model ($p < 0.05$, Holm-Bonferroni, Appendix Table 18).

| Model | ABIDE | HBN CELF | HBN WISC | HBN Age | HBN Sex | UKB Age | UKB Sex | SRPBS MDD | SRPBS SZ |
|---|---|---|---|---|---|---|---|---|---|
| BrainLM | 53.84† ± 3.00 | 42.03† ± 3.41 | 38.26 ± 4.11 | 43.89† ± 2.12 | 65.44† ± 1.72 | 30.16 ± 1.41 | 86.71† ± 0.63 | 57.61† ± 4.14 | 55.72† ± 6.62 |
| Brain-JEPA | 52.92† ± 3.53 | 41.50 ± 5.16 | 38.34 ± 3.42 | 39.81† ± 2.08 | 63.96† ± 1.45 | 30.60 ± 2.12 | 83.23† ± 1.26 | 52.72† ± 4.18 | 57.63† ± 3.75 |
| Brain-Semantoks | **65.13** ± 2.14 | **42.18** ± 2.80 | **40.87** ± 2.43 | 39.16† ± 0.81 | **69.52** ± 0.93 | **31.15** ± 1.15 | **87.52** ± 0.52 | **62.60** ± 4.79 | **69.26** ± 3.98 |

Table 2: Comparisons with supervised and finetuned baselines (Balanced Accuracy (%)). Best results in **bold**, second-best underlined. †Significantly worse than best model ($p < 0.05$, Holm-Bonferroni, Appendix Table 19).

| Model | UKB-Age | UKB-Sex | HBN-Age | HBN-Sex | HBN-CELF | HBN-WISC |
|---|---|---|---|---|---|---|
| FC | 27.04† ± 1.51 | 80.63† ± 0.89 | 41.81† ± 1.36 | 66.51† ± 1.42 | 42.41 ± 2.91 | 39.79 ± 2.91 |
| BNT | 20.48† ± 1.08 | 77.91† ± 3.37 | 22.59† ± 4.31 | 61.74† ± 9.69 | 42.40 ± 3.98 | 38.53 ± 2.94 |
| BolT | 26.85† ± 1.59 | 80.30† ± 0.91 | 37.67† ± 1.41 | 65.22† ± 1.23 | 42.45 ± 1.78 | 39.53 ± 4.42 |
| BrainMass | 23.51† ± 1.69 | 69.72† ± 1.89 | 31.80† ± 1.01 | 56.97† ± 1.08 | 36.93† ± 2.06 | 38.00 ± 2.83 |
| BrainLM | 30.26† ± 1.66 | 85.75† ± 0.77 | 39.31† ± 2.02 | 64.37† ± 2.32 | 39.27 ± 4.46 | 35.34 ± 3.61 |
| Brain-JEPA | 30.60† ± 1.60 | 86.70† ± 1.20 | **41.91** ± 2.00 | 65.57† ± 2.28 | 39.60† ± 3.50 | 35.20† ± 3.10 |
| Brain-Semantoks | 31.15† ± 1.15 | **87.52** ± 0.52 | 39.16 ± 0.81 | **69.52** ± 0.93 | 42.18 ± 2.80 | **40.87** ± 2.43 |
| + Finetune | **33.91** ± 0.87 | 87.13 ± 0.57 | 39.41 ± 3.77 | 69.31 ± 0.71 | **42.59** ± 1.34 | 40.82 ± 1.51 |

| Model | LEMON-CVLT | LEMON-MDBF | LEMON-TMT | ABIDE | SRPBS-MDD | SRPBS-SZ | Average |
|---|---|---|---|---|---|---|---|
| FC | 39.49 ± 8.32 | 32.29 ± 6.09 | 41.14 ± 6.58 | 65.12 ± 2.98 | 60.30 ± 4.65 | **71.59** ± 5.84 | 50.68† |
| BNT | 36.76† ± 4.90 | 37.90 ± 5.12 | 33.86† ± 6.13 | 58.38 ± 6.51 | 57.60† ± 4.57 | 66.59† ± 5.09 | 46.23† |
| BolT | 39.54 ± 4.71 | 37.98 ± 5.82 | 40.30 ± 7.52 | 64.89 ± 4.08 | 59.50† ± 4.52 | 67.12 ± 6.23 | 50.11† |
| BrainMass | 32.24† ± 3.10 | 31.96† ± 5.21 | 39.14† ± 7.00 | 60.89† ± 3.68 | 59.82† ± 4.45 | 70.22† ± 6.20 | 48.43† |
| BrainLM | 37.81 ± 6.91 | 34.05 ± 1.84 | 35.33† ± 3.78 | 53.91† ± 2.23 | 54.29† ± 2.38 | 60.10† ± 5.79 | 47.48† |
| Brain-JEPA | 30.94† ± 6.20 | 32.26† ± 6.58 | 35.48† ± 9.58 | 52.20† ± 4.00 | 54.00† ± 4.00 | 60.50† ± 4.40 | 47.08† |
| Brain-Semantoks | 42.10 ± 4.72 | **40.23** ± 5.74 | **42.88** ± 4.05 | 65.13 ± 2.14 | 62.60 ± 4.79 | 69.26 ± 3.98 | 52.72 |
| + Finetune | **44.36** ± 3.36 | 38.58 ± 4.65 | 39.21 ± 1.91 | **65.44** ± 1.16 | **63.60** ± 2.56 | 71.05 ± 4.39 | **52.95** |

**Evaluation**

We evaluate representations in two settings: linear probing and full fine-tuning. For linear probing, we freeze the pretrained teacher encoder and train a single linear layer on top of its outputs to assess raw representation quality. For fine-tuning, the entire model is updated to provide a comparison with supervised baselines. All evaluations use ten cross-validation repetitions while stratifying on the target. At test time, we average predictions of 8 equally-spaced temporal crops for all methods. For Brain-Semantoks evaluations and ablations, we pretrain using three random seeds and average their scores for each CV iteration to improve reliability.

The linear probing setup is standardized across models (Appendix Table 17). For Brain-Semantoks, the input to the linear layer is the concatenation of the teacher's `[CLS]` token and the average of all network-patch tokens. For the linear probing comparisons, we omit the LEMON dataset as we only have access to filtered, preprocessed data. The ROI-specific offsets that BrainLM and Brain-JEPA were pretrained on are therefore not present, and we observe chance-level performance of these models on this dataset. We note that the normalization strategy used for Brain-Semantoks is more universally applicable however.

The two main baselines we compare against are BrainLM (Caro et al., 2023) and Brain-JEPA (Dong et al., 2024), which are both fMRI time series foundation models using parcellated data. Importantly, both models are pretrained solely on the UKB, which is necessary for informative comparisons. Please see Appendix A.1 for further baseline model descriptions.

# 5 RESULTS

We evaluated Brain-Semantoks on a diverse set of downstream tasks, assessing its performance via linear probing against state-of-the-art foundation models and fully supervised methods. We then conducted extensive scaling and ablation studies to validate our architectural and training choices, and finally leveraged our model's unique properties for built-in interpretability.

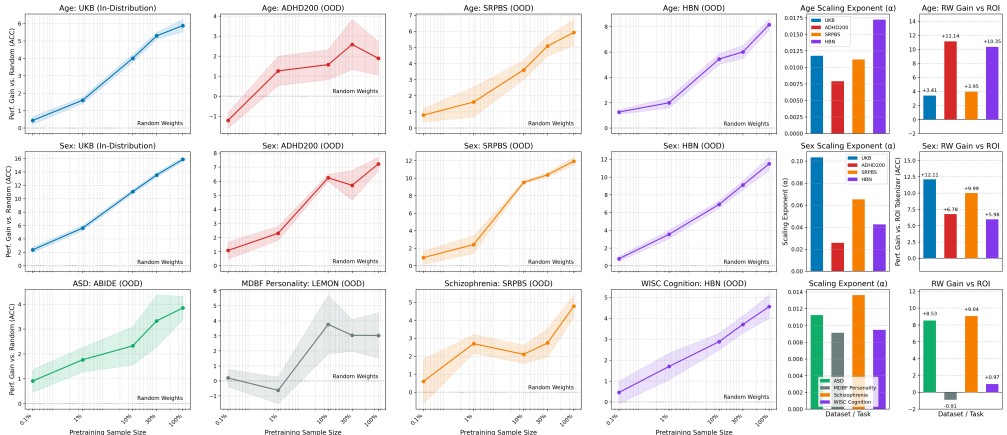

Figure 2: Scaling performance of linear probing following pretraining on increasing sample sizes. We compare within and out-of-distribution scaling. RW: Random weights. ROI: We compare to using a linear layer for projecting single-ROI timeseries instead of our semantic tokenizer.

Table 3: Task-based fMRI prediction using balanced accuracy (%) on the Hariri emotion task.

| Model | Linear Probing | | | Finetuning | | |
|---|---|---|---|---|---|---|
| | 1 Block (Pad) | 2 Blocks (Cont.) | 2 Blocks (Cat.) | 1 Block (Pad) | 2 Blocks (Cont.) | 2 Blocks (Cat.) |
| Brain-JEPA | 81.45 ± 0.59 | 82.29 ± 0.28 | 81.06 ± 0.81 | 91.04 ± 1.56 | 92.33 ± 1.86 | 94.71 ± 0.85 |
| Brain-Semantoks (Ours) | **93.84 ± 0.36** | **94.34 ± 0.75** | **96.50 ± 0.15** | **96.89 ± 0.73** | **97.85 ± 0.86** | **97.70 ± 0.80** |

## 5.1 DOWNSTREAM PERFORMANCE

A primary goal for a foundation model is to produce representations that are directly useful for downstream tasks without extensive fine-tuning. We therefore prioritize evaluation using a rigorous linear probing protocol, where the pretrained model weights are frozen.

As shown in Table 1, Brain-Semantoks consistently and significantly outperforms existing fMRI foundation models, which are based on reconstruction objectives. Our model achieves the highest accuracy on eight of the nine tasks, often by a large margin. The improvements are particularly striking on challenging out-of-distribution clinical datasets, where Brain-Semantoks achieves large performance gains for predicting ASD, Schizophrenia, and MDD.

We next compared the linear probing performance of Brain-Semantoks to fully fine-tuned models and strong supervised baselines in Table 2. With only a linear probe, Brain-Semantoks outperforms all baselines on eight diverse tasks. The ability to surpass fully supervised models, which are trained end-to-end on task-specific data, highlights the utility of the learned representations.

Finally, we evaluate generalization to task-based fMRI using the Hariri emotion task from UKB, where participants match shapes or emotional faces in blocks. We predict block type from short segments, requiring within-subject discrimination across substantially shorter timescales than pretraining sequences. We address this temporal mismatch by leveraging the masked distillation framework. Specifically, we construct a single temporal patch from a recording segment (e.g. a task block) and mask all remaining positions, thereby requiring the model to generate summary representations from limited context. We explore multiple patch construction strategies which we visualize and detail in Appendix A.7. Brain-Semantoks substantially outperforms Brain-JEPA across all settings (Table 3).

## 5.2 SCALING LAWS OF SEMANTIC FMRI REPRESENTATIONS

We conducted the first detailed scaling analysis for fMRI foundation models under a linear probing protocol to understand how performance varies with pretraining data size. We trained Brain-Semantoks on subsets of the UKB dataset and evaluated on both in-distribution (UKB hold-out) and out-of-distribution tasks.

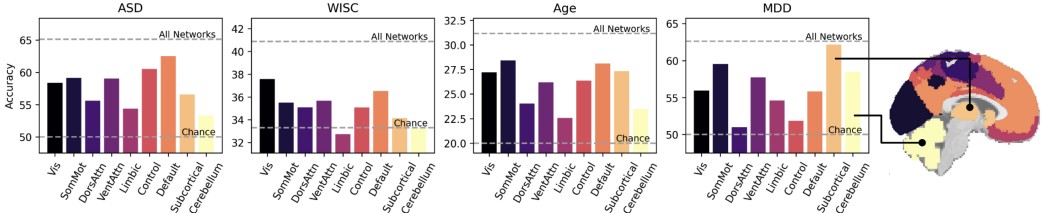

Figure 3: We investigate the predictive performance of individual network representations.

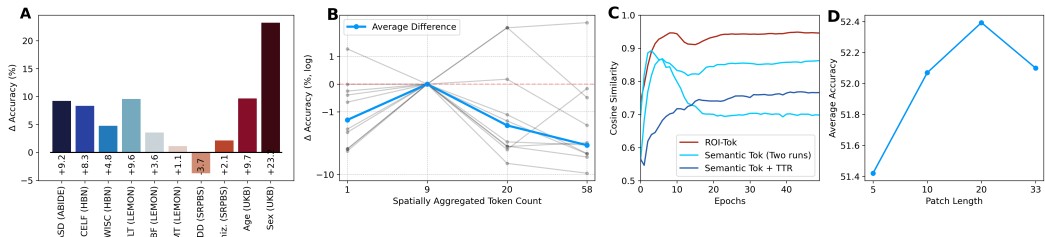

Figure 4: Tokenizer Ablations. A) Linear probing comparison between our semantic tokenizer vs (minus) a linear projection layer applied to ROIs individually. B) We ablate aggregating into different numbers of 'networks' or spatial tokens. Grey lines denote downstream tasks. C) Pretraining dynamics visualised by the cosine similarity between the teacher tokens and reconstructed student tokens. D) Temporal patch size ablation. TTR: Teacher-guided Temporal Regularizer

As shown in Figure 2, performance on nearly all tasks improves predictably with the logarithm of the pretraining data size, following a power-law relationship characteristic of foundation models in other domains. Critically, we observe strong scaling laws for out-of-distribution (OOD) generalization. For age and sex prediction ($n_{train,probe} = 500$), which enable comparisons for matched prediction tasks, we observe consistent performance increases with more pretraining data. Remarkably, we note strong and reliable scaling for HBN, which has an age gap of more than 20 years with the UKB (HBN: up to 22, UKB: from 44 years old). Across the majority of tasks, we observe no plateau in OOD scaling. Yet, in contrast to fields such as language processing, downstream probing performance is not only affected by scaling, but also by the baseline performance. We found that this starting point is significantly increased in Brain-Semantoks, as it outperforms ROI-level projection by up to 12% when both are randomly initialized.

## 5.3 INTERPRETABILITY

A common challenge for interpreting deep models is that post-hoc analyses, like input masking, shift the data out of the training distribution, potentially yielding unreliable results. Our distillation pretraining with slice-masking directly addresses this. Because the model is trained to predict global information from seeing only a subset of brain networks, we can probe its learned dependencies in an "in-distribution" manner.

We assessed the importance of each of the 9 functional networks for various downstream tasks by masking all but one network and evaluating linear probing performance (Figure 3). This reveals which individual networks contain the most predictive information for a given phenotype. Multiple findings align well with neuroscientific research such as the importance of the default-mode network for ASD and subcortical regions for MDD (Ramasubbu et al., 2014). Interestingly, whereas the default-mode network has dominated MDD research, we find that cerebellar activity is more predictive, which is a more recent hypothesis (Wang et al., 2023).

## 6 ABLATIONS

We perform post-hoc ablation studies on core aspects of the Brain-Semantoks framework to analyze the contribution of each component. We average linear probing performance across ten downstream

Table 4: Tokenizer Conv.

| Kernel | Score |
|---|---|
| *Single Branch* | |
| Dense 20 | 48.28 |
| Dense 3 | 50.50 |
| DW-Str 16 | 52.19 |
| *Dual Branch* | |
| Dense 3 + DW 16 | 51.35 |
| **Dense 3 + DW-Str 16** | **52.39** |

Table 5: TTR Duration

| Duration | Score |
|---|---|
| 0% | 50.88 |
| **5%** | **52.39** |
| 100% | 49.60 |

Table 6: Masking Type

| Masking Type | Score |
|---|---|
| Random | 51.03 |
| Block | 49.71 |
| Network | 52.09 |
| Temporal Slice | 51.50 |
| **Slice** | **52.39** |

Table 7: Loss Components

| CLS | $\lambda_{Tok}$ | Score |
|---|---|---|
| No | 1.0 | 47.32 |
| Yes | 0.0 | 50.10 |
| **Yes** | **0.5** | **52.39** |
| Yes | 1.0 | 51.62 |

Table 8: Masking Ratio

| Ratio | Score |
|---|---|
| [0.1, 0.9] | 50.33 |
| [0.45, 0.55] | 51.23 |
| [0.5, 0.75] | 51.88 |
| **[0.65, 0.85]** | **52.39** |

Ablation studies using average balanced accuracy for linear probing across ten downstream tasks. In table 4: 'Dense' and 'DW' denote standard (channel-mixing) and depthwise convolutions, respectively. Suffix '-Str' indicates the proposed structured kernel (as opposed to standard free-form). Integers denote kernel length. $\lambda_{Tok}$: Weight on token masking loss. TTR Duration: active pretraining period. *Loss Components*: First row (CLS=No) represents JEPA-style baseline.

tasks, omitting HBN-Age and HBN-Sex to reduce the influence of demographic prediction on the total score. First, we compare using a linear projection layer operating on single ROIs (as in Brain-JEPA), instead of our semantic tokenizer (Figure 4A). We observe unstable training dynamics, as the cosine similarity between the student's reconstructed tokens and teacher tokens (i.e., the negative of the token-level loss) to quickly reach 0.95 and stabilize there (Figure 4C). This indicates partial collapse where the learned representations are simple and can be predicted at high accuracy immediately, which is associated with poor downstream performance. Indeed, we find large gains in downstream performance by adopting the semantic tokenizer, which also results in improved training dynamics, although instability still exists early in training. By including Teacher-guided Temporal Regularisation (TTR), which we decay to zero in the first 5% of training, we observe stable pretraining dynamics and strong downstream performance. We find using TTR for the entirety of pretraining to be overly restrictive (Table 5). We also ablate the global distillation loss (CLS Loss in Table 7), which we observe to be important for downstream performance.

We furthermore ablate the choice of nine functional networks for the semantic tokenizer and we compare to spatially aggregating more aggressively (into 1 spatial token per temporal patch) or less so (20 or 58 spatial tokens; Figure 4B; see appendix Table 13 for details). We observe a significant bias for most downstream tasks towards fewer spatial tokens, with the best overall performance for the nine network solution. We also ablate the temporal patch size and convolutional filter bank for the tokenizer, finding that relatively long patches and structured kernels are important (Table 4).

Next, we provide ablations on masking. We find that the influence of the mask loss should not be too high (Table 7), masking types which reduce interpolation learning are most effective (Table 6), combined with a high masking ratio (Table 8). Finally, we perform ablations on atlas choice (Appendix A.4.1), temporal resolution (Appendix A.4.2), augmentations (Appendix A.4.3), crop length $T_{crop}$ (Appendix A.4.4), and the coding rate regularizer $R_\epsilon$ (Appendix A.4.5).

# 7 DISCUSSION

This paper presents Brain-Semantoks, a novel foundation model for fMRI that marks a significant shift to learning abstract, semantic representations of brain dynamics. By introducing a neuroscientifically-grounded semantic tokenizer and employing a self-distillation objective, the model effectively learns high-level phenotypic signatures. The results demonstrate the strength of this approach, achieving state-of-the-art performance under a rigorous linear probing protocol and often surpassing supervised methods on diverse tasks. This indicates the learned representations are broadly applicable without domain adaptation.

Future work may benefit from more extensive integration of task-based fMRI data and thorough investigations to understand which distribution shifts are particularly harmful for transfer. Furthermore, while we find that using neuroscience-based functional networks is effective for many downstream tasks, follow-up research may explore learning how to group ROIs from data rather than relying on fixed mappings.

ACKNOWLEDGMENTS

This research was funded by Gemeinnützigen Hertie-Stiftung and the Deutsche Forschungsgemeinschaft (DFG) through FOR 5187 (project number 442075332). Additional support was provided by the Machine Excellence Cluster and DFG through the Germany's Excellence Strategy (EXC 2064 - project number 390727645) and the following projects: CRC 1404 (project number 414984028), TRR 265 (project number 402170461), and RU 5363 (project number 459422098).

ETHICS STATEMENT

This research utilized publicly available, pre-existing neuroimaging datasets, including the UK Biobank, ABIDE, HBN, SRPBS, LEMON, and ADHD200. No new data were collected for this study. All data were acquired in accordance with the data use agreements specific to each dataset, such as the UK Biobank application process. The original data collection procedures for these cohorts were conducted under the approval of their respective institutional review boards (IRBs) and ethics committees, with all participants providing informed consent. We have complied with all data privacy and sharing agreements, ensuring that sensitive information is stored securely and handled according to the established protocols.

REPRODUCIBILITY STATEMENT

To ensure the reproducibility of our findings, we provide code and pretrained models at `https://github.com/SamGijsen/Brain-Semantoks`. A comprehensive description of our methodology, including model architecture, training procedures, and evaluation protocols, is provided in the Experimental Setup (Section 3). Specific hyperparameter configurations for all stages of our experiments are detailed in Appendix Table 17. All datasets used in this work are publicly accessible, and relevant citations are provided to guide their acquisition. These resources are intended to allow for the full verification of our results and facilitate future research building upon our work.

LLM USAGE STATEMENT

The authors utilized large language models (LLMs) during the preparation of this manuscript. Specifically, Google's Gemini was used for proofreading, correcting grammatical errors, and improving sentence structure for clarity. All LLM-generated suggestions were manually reviewed and edited by the authors to ensure the final text accurately reflects our research and claims. The core scientific contributions and claims were solely authored by the human authors.

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

## A APPENDIX

### A.1 BASELINE METHODS

Besides BrainLM and Brain-JEPA, we further compare against strong supervised approaches. The most widely-used method for prediction with fMRI is to compute pairwise Pearson correlations between ROI time series as a measure of 'functional connectivity' (FC) and use a support vector machine for prediction. We select the regularisation strength using the validation set ($C = \{10^{-3}, 10^{-2}, 10^{-1}, 1, 10, 10^2, 10^3\}$). Additionally, BolT is a Transformer-based architecture that classifies fMRI time series by using a novel fused window attention mechanism to hierarchically build representations from local to global temporal scales (Bedel et al., 2023). Next, Brain Network Transformer (BNT) utilizes ROI connection profiles for positional encoding and introduces an Orthonormal Clustering Readout mechanism. This novel pooling function learns cluster-aware embeddings by grouping functionally similar brain regions to improve graph-level predictions (Kan et al., 2022). Finally, BrainMass operates on functional connectivity values and uses multi-task pretraining to learn a latent representation robust to random masking (Yang et al., 2024).

### A.2 EXTENDED TASK INFORMATION

- **HBN: WISC FSIQ.** The Wechsler Intelligence Scale for Children (Wechsler, 1949) provides a broad measure for an individuals cognitive ability (i.e., 'Full Scale Intelligence Quotient'), spanning verbal comprehension, visual spatial reasoning, fluid reasoning, working memory, and processing speed.

- **HBN: CELF.** The Clinical Evaluation of Language Fundamentals test provides a measure of multi-dimensional language performance, including language memory, language structure, language context, language expressivity, and language receptivity (Semel et al., 2003). We use the total score of the scale.

- **LEMON: CVLT.** A German adaptation of the California Verbal Learning Test (Delis et al., 2008), which is a memory-based task assessing how well a person can recall and recognize a list of words.

- **LEMON: TMT.** The Trail Making Test is a widely used neuropsychological test assessing cognitive functions (Reitan, 1992). We use the commonly used 'B-A' score, which is a difference score between a simpler (A) and complex (B) task, which aims to isolate executive functioning.

- **LEMON: MDBF.** A mood state questionnaire administered in German (Mehrdimensionale Befindlichkeitsfragebogen; Steyer et al. (1997)) The three subscales (Good-Bad, Awake-Tired, Calm-Nervous) were averaged to create a composite score of general positive affective state, following observation of strong inter-correlations between the components ($r = .51$-$.71$).

Table 9: Summary of datasets, tasks, and sample sizes for downstream evaluations. "Tr/V/T" stands for Train/Validation/Test.

| Dataset | Task | Classes | Total Size | Split Ratio (Tr/V/T) | Train Size |
|---------|------|---------|-----------|---------------------|-----------|
| UKB | Sex | 2 | 1625 | 0.31 / 0.08 / 0.61 | 500 |
| UKB | Age | 5 | 1625 | 0.31 / 0.08 / 0.61 | 500 |
| HBN | Sex | 2 | 1870 | 0.27 / 0.27 / 0.46 | 500 |
| HBN | Age | 5 | 1870 | 0.11 / 0.11 / 0.78 | 200 |
| HBN | WISC FSIQ | 3 | 884 | 0.60 / 0.20 / 0.20 | 530 |
| HBN | CELF Total | 3 | 1005 | 0.60 / 0.20 / 0.20 | 603 |
| LEMON | CVLT | 3 | 212 | 0.60 / 0.20 / 0.20 | 127 |
| LEMON | TMT B-A | 3 | 212 | 0.60 / 0.20 / 0.20 | 127 |
| LEMON | MDBF | 3 | 213 | 0.60 / 0.20 / 0.20 | 127 |
| SRPBS | Schizophrenia | 2 | 291 | 0.60 / 0.20 / 0.20 | 174 |
| SRPBS | MDD | 2 | 499 | 0.60 / 0.20 / 0.20 | 299 |
| ABIDE | Autism | 2 | 974 | 0.60 / 0.20 / 0.20 | 584 |

## A.3 TOKENIZER CONVOLUTIONS

The $\text{Conv}_{\text{std}}$ branch uses short kernels for local, cross-ROI dependencies. Meanwhile, the $\text{Conv}_{\text{str}}$ branch uses longer, decaying kernels to enforce a temporal inductive bias absent in standard ViT architectures. Specifically, due to the hemodynamic properties of the BOLD signal, it is both highly autocorrelated and smooth. We therefore hypothesize a temporal locality prior is beneficial, combined with a longer kernel to model the slow-evolving signal.

To transform the input time series from $C_n$ ROIs to $\frac{D}{2}$ channels, an expansion factor is computed as $H = \text{ceil}\left(\frac{D/2}{C_n}\right)$ ('heads'). Each ROI is then processed by its set of $H$ filters, resulting in $H \times C_n$ total channels. We furthermore adopt a learnable residual connection as used in (Li et al., 2022; Vetter et al., 2024), which is analogous to the direct feedthrough term used in state-space models. Finally, a linear projection layer maps from $H \times C_n$ to the target $\frac{D}{2}$ dimensionality, ensuring a fixed output dimensionality while also mixing information across ROIs.

Whereas the standard convolutions ($\text{Conv}_{\text{std}}$) uses a kernel size of 3, the structured, depthwise-separable convolutions ($\text{Conv}_{\text{str}}$) use a base kernel size of 4 across 3 scales. Specifically, a kernel of length 16 is created by concatenating three kernels: a 4-length kernel with weight 4.0, a 4-length kernel with weight 2.0, and an 8-length kernel (upsampled from 4 parameters) with weight 1.0.

## A.4 EXTRA ABLATION EXPERIMENTS

### A.4.1 ABLATIONS OF BRAIN ATLASES

We present ablation results on the atlas (parcellation) choices in Table 10. Overall, these ablations are in line with the literature in that a moderate amount of ROIs generally works well. If too few are used, the parcellation step destroys too much signal. Too many and one is liable to model too much noise. We note that for the Gordon-333 atlas (Gordon et al., 2016) we map the 333 cortical ROIs to 13 'communities' using the provided membership; the parcellation thereby offers an alternative aggregation solution to the Yeo-networks. Specifically, these communities

| Atlas Combination | ASD ABIDE | CELF HBN | WISC HBN | CVLT LEMON | MDBF LEMON | TMT LEMON | MDD SRPBS | SZ SRPBS | Age UKB | Sex UKB | Global Avg |
|---|---|---|---|---|---|---|---|---|---|---|---|
| Base (S400-T3-B7) | 65.13 | 42.18 | 40.87 | 42.10 | 40.23 | 42.88 | 62.60 | 69.26 | 31.15 | 87.52 | 52.39 |
| *Cortical Parcellation Variations* | | | | | | | | | | | |
| S400 → S100 | -1.82 | -2.74 | -1.49 | -2.19 | -4.61 | -3.91 | +0.03 | +0.78 | -1.75 | -8.02 | -2.57 |
| S400 → S800 | -0.28 | +0.56 | -1.89 | -2.33 | -3.16 | -3.07 | -1.27 | -2.14 | -0.16 | +0.41 | -1.33 |
| S400 → Gordon333 | -1.41 | +1.44 | +0.15 | +1.52 | -1.90 | +0.10 | -2.60 | -0.85 | -2.70 | -2.88 | -0.91 |
| *Subcortical Parcellation Variation* | | | | | | | | | | | |
| T3 → T1 | -0.46 | -0.48 | -0.47 | -0.16 | -2.01 | -2.86 | -1.70 | -2.10 | -0.30 | -0.69 | -1.12 |
| *Cerebellar Parcellation Variation* | | | | | | | | | | | |
| B7 → B17 | -0.27 | +0.76 | -1.48 | +2.36 | -0.06 | -0.04 | -2.17 | -0.17 | -0.14 | -0.09 | -0.13 |

Table 10: Performance comparison across atlas variations presented as average balanced accuracy (%) across three model seeds, each evaluated using 10 cross-validation repetitions. Base case uses Schaefer-400, Tian-3, Buckner-7. Changes are absolute differences in balanced accuracy (percentage points) relative to base.

emerged from applying Infomap, a data-driven community detection algorithm, to parcel-to-parcel connectivity data. The community names are as follows: Default, SomatomotorHand, SomatomotorMouth, Visual, Frontoparietal, Auditory, CinguloParietal, RetrosplenialTemporal, CinguloOpercular, Salience, DorsalAttention, and remaining ROIs. Interestingly, the Gordon-333 solution with a total of 15 spatial tokens still yields strong downstream performance, indicating robustness to the specific network mapping that is chosen.

| Repetition Time | ASD ABIDE | CELF HBN | WISC HBN | CVLT LEMON | MDBF LEMON | TMT LEMON | MDD SRPBS | SZ SRPBS | Age UKB | Sex UKB | Global Avg |
|---|---|---|---|---|---|---|---|---|---|---|---|
| Base (2 sec, 0.5Hz) | 65.13 | 42.18 | 40.87 | 42.10 | 40.23 | 42.88 | 62.60 | 69.26 | 31.15 | 87.52 | 52.39 |
| *Repetition Time Variations* | | | | | | | | | | | |
| 2 → 0.735 sec (1.36Hz) | -6.45 | -0.15 | -0.80 | +0.84 | -2.21 | +2.62 | -2.50 | -2.42 | -0.61 | -0.25 | -1.19 |

Table 11: Performance comparison across repetition time variations. Base case uses 2 sec (0.5 Hz). Changes are absolute differences in balanced accuracy (percentage points) relative to base. Global average computed as mean across the 10 tasks.

### A.4.2 ABLATION OF TEMPORAL RESOLUTION

We present ablation results on the temporal resolution (repetition time; TR) in Table 11. For this analysis, we did not resample the UKB data but kept it at its native TR of 0.735s for pretraining. All downstream data was upsampled to this resolution. To accommodate the longer sequences, we increase the patch length and crop length $T_{crop}$ by a factor of $\approx \frac{2.0}{0.735}$. We furthermore increase the structured convolution kernel length by increasing the base kernel from 4 to 6 and the number of scales from 3 to 4. We observe that particularly for disease classification (ASD, MDD, SZ) performance drops are most severe. While this may relate to the nature of the downstream tasks, it is likely that the relatively high TR of their datasets played a significant role. Indeed, the SRPBS dataset has TRs of 2-2.5s and although ABIDE has a wider range, most are in the 2-3s range. For comparison, HBN has a TR of 0.8s, LEMON of 1.4s, and the UKB of 0.735s. The largest gains are seen for the prediction of cognition (CVLT and TMT) on the LEMON dataset, indicating these phenotypes may benefit from a higher temporal resolution. In sum, we believe that pretraining on resampled data (e.g. with a TR of 2s) is sensible if one aims to evaluate on downstream data which is generally also resampled and has a relatively higher native TR.

### A.4.3 ABLATION OF DATA AUGMENTATIONS

To investigate the role of data augmentations during pretraining, we compare physiologically-motivated augmentations against the corruption-based augmentations used in our main experiments. We test three augmentation strategies designed to mimic fMRI artifacts and noise sources:

**ROI Mixing.** Simulates cross-ROI signal interference (e.g., from head motion or spatial blurring) by blending signals between spatially adjacent ROIs. For a fraction $p_{\text{rois}}$ of ROIs, we select a random

neighbor from the anatomical adjacency matrix and apply a convex combination:

$$\text{signal[roi]} = \alpha \cdot \text{signal[roi]} + (1 - \alpha) \cdot \text{signal[neighbor]} \tag{7}$$

where $\alpha \sim \mathcal{U}(0.7, 0.95)$. We test light ($p_{\text{rois}} = 0.1$) and heavy ($p_{\text{rois}} = 0.2$) variants.

**Frequency Masking.** Attenuates power in the typical BOLD frequency range (0.01–0.1 Hz) via FFT. For each ROI, we mask a random subset of frequency bins (with probability $\sim \mathcal{U}(0.1, 0.3)$ per ROI) by scaling with factors $\sim \mathcal{U}(s_{\text{min}}, 1.0)$, then reconstruct via inverse FFT. We test $s_{\text{min}} = 0.8$ (light) and $s_{\text{min}} = 0.6$ (heavy).

**Band-Specific Noise.** Injects synthetic physiological noise as 1–2 sinusoidal components per ROI in the 0.01–0.1 Hz band:

$$\text{noise} = \sum_i a_i \sin(2\pi f_i t + \phi_i) \tag{8}$$

where amplitudes $a_i \sim \mathcal{U}(0.5, 1.5) \cdot \sigma_{\text{noise}}$, frequencies $f_i \sim \mathcal{U}(0.01, 0.1)$, phases $\phi_i \sim \mathcal{U}(0, 2\pi)$, and $\sigma_{\text{noise}} \sim \mathcal{U}(0, \sigma_{\text{max}})$. We test $\sigma_{\text{max}} = 0.1$ (light) and $\sigma_{\text{max}} = 0.2$ (heavy).

We pretrain models using: (1) temporal views only (no further augmentation), (2) physiologically-motivated augmentations at light and heavy intensities, and (3) corruption-based augmentations (as in main paper). All models are evaluated on the same 10 downstream tasks. Results are shown in Table 12.

Table 12: Ablation of data augmentation strategies. Performance reported as average accuracy across 10 downstream tasks.

| Augmentation Strategy | Avg. Accuracy (%) |
|---|---|
| Temporal views only | 51.91 |
| + Physiological (light), or | 51.85 |
| + Physiological (heavy), or | 51.92 |
| + Corruption (main paper) | **52.39** |

We find that physiologically-motivated augmentations provide no benefit over temporal view selection alone, while simple corruption augmentations yield a small improvement. We hypothesize that (1) fMRI data already contains substantial physiological noise, so our cross-view distillation objective implicitly learns invariance to these factors without explicit augmentation. (2) Our input data follows a conventional bandpass filter (0.01-0.1Hz) to isolate hemodynamic responses. Consequently, we were constrained to applying frequency-based augmentations within this signal-rich band. This likely creates a destructive trade-off: attempting to simulate spectral noise in this range and introduce data diversity risks corrupting the underlying neural signal. Similarly, we excluded global spike augmentations (as an alternative to mimic head motion effects) as such artifacts are inconsistent with bandpass-filtered inputs. The effectiveness of corruption-based augmentations may stem from their ability to increase data diversity without introducing structured artifacts that could interfere with the semantic-level representations our tokenizer targets.

Table 13: Ablation of Spatial Token Aggregation Strategies

| Spatial Tokens | Aggregation Strategy |
|---|---|
| 1 | All 457 ROIs are aggregated into a single group. |
| 9 | 7 Yeo networks + 1 subcortical group + 1 cerebellum group. |
| 20 | 17 Yeo networks + 2 manually split subcortical groups + 1 cerebellum group. |
| 58 | 17 Yeo networks (each split 3 times) + 6 manually split subcortical groups + 1 cerebellum group.* |

We always use the Schaefer-400, Tian-3, and Buckner-7 parcellations. We used 'slice masking' except when using a single spatial token, as there it is only possible to mask temporal slices. We standardized the creation of masks as follows: masks were initially sampled on the level of the nine spatial tokens and, when using 20 or 58 spatial tokens, upsampled to the required spatial 'resolution'. Directly sampling masks with higher resolutions would have allowed for simpler interpolation solutions during pretraining, potentially biasing the analysis towards fewer spatial tokens.
*Note: The cerebellum parcellation was not split as the atlas contains only 7 ROIs for this region.

### A.4.4 ABLATION ON $T_{\text{CROP}}$

We perform a post-hoc analysis on the sensitivity to $T_{\text{crop}}$ by pretraining with crop lengths of 80, 100, and 120 timepoints and measuring average performance across our 10 downstream tasks. We observe that performance does not increase monotonically with crop length. While longer crops provide more information per sample (and we indeed find single-crop prediction improves: 50.58% $\rightarrow$ 51.33% $\rightarrow$ 51.60%, while we average over the default 8 crops in Table 14), we hypothesize this is offset by several factors:

- Distillation task difficulty: Longer crops make the student-teacher matching task easier, reducing pressure to learn strong, generalizable representations.
- Data diversity: Longer crops reduce the number of distinct crops per recording, decreasing sample-level diversity during pretraining.
- Ensembling benefits: Longer crops have greater overlap, reducing the benefit of multi-crop ensembling during inference.

Our default choice of $T_{\text{crop}} = 100$ appears to represent a favorable balance between information content and the above pretraining considerations. Overall, the method is reasonably robust across this range.

| $T_{\text{crop}}$ | ASD ABIDE | CELF HBN | WISC HBN | CVLT LEMON | MDBF LEMON | TMT LEMON | MDD SRPBS | SZ SRPBS | Age UKB | Sex UKB | Global Avg |
|---|---|---|---|---|---|---|---|---|---|---|---|
| Base (100) | 65.13 | 42.18 | 40.87 | 42.10 | 40.23 | 42.88 | 62.60 | 69.26 | 31.15 | 87.52 | 52.39 |
| *$T_{\text{crop}}$ Variations* | | | | | | | | | | | |
| 80 | -0.05 | +0.17 | -0.48 | -0.79 | -2.07 | -0.12 | -1.67 | -1.86 | +0.28 | -0.33 | -0.69 |
| 120 | +0.62 | -0.28 | +1.03 | +0.38 | -1.36 | -1.00 | -0.03 | +0.35 | +0.17 | -0.41 | -0.15 |

Table 14: Performance comparison across $T_{\text{crop}}$ variations. Changes are absolute differences in balanced accuracy (percentage points) relative to base. Global average computed as mean across the 10 tasks.

### A.4.5 ABLATION ON CODING RATE REGULARIZER $R_\epsilon$

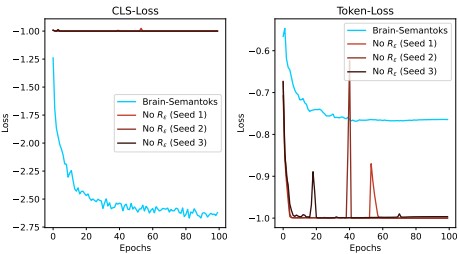

Figure 5: We observe model collapse at the start of training when the coding rate regularizer is ommitted.

### A.5 OOD ROBUSTNESS ANALYSIS

We perform additional robustness analyses pertaining to out-of-distribution generalization. Specifically, we create data subsets based on site, MRI scanner, repetition time (TR), spatial resolution, and field-of-view (FOV). For both SRPBS and ABIDE we perform sex and age prediction to maximize sample size availability. We probe Brain-Semantoks, as well as versions trained without the global CLS loss or without the Semantic Tokenizer (i.e., using ROI-wise linear projection). For SRPBS, the two ablated models exhibit stronger performance drops for increased TR, lowered resolution, and lowered FOV.

Overall, results indicate that both components are important for consistent performance. However, it is not obvious from these results that certain investigated scanning parameters are particularly

problematic for transfer learning. Although these analyses are not exhaustive and therefore do not preclude the possibility that other factors would provide critical insight, these findings suggest uniform robustness improvements from the complete model. Such patterns may result from generally improved representation quality that is robust to scanning parameter variations rather than mere overfitting to pretraining data characteristics.

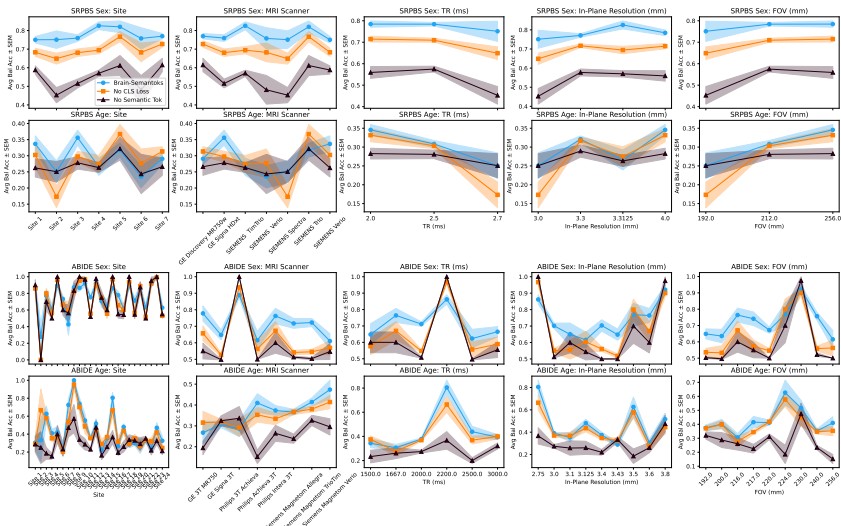

Figure 6: OOD Robustness Analyses. Three model seeds are evaluated using ten cross-validation repetitions. The mean and SEM of downstream performance is plotted. We compare Brain-Semantoks (blue) to the ablated versions trained without CLS-loss (orange) or without Semantic Tokenizer (black). For reference, the data used for pretraining was acquired on a Siemens Skyra scanner with TR=0.735s, an in-plane resolution of 2.4mm, and a FOV of 211mm. Semantic Tok: Semantic Tokenizer.

## A.6 DETAILS ON FMRI FOUNDATION MODEL COMPARISONS

To provide a fair comparison between fMRI foundation models, we standardized the linear probing evaluations and in doing so discovered significant effects on performance. This led to much stronger BrainLM evaluations than previously reported (Dong et al., 2024). First, the temporal crop or window size for BrainLM ($\tilde{2}$.5 minutes) is shorter than Brain-Semantoks ($\tilde{3}$.3 minutes) and BrainJEPA (almost 6 minutes). As these models are mainly used for subject-level prediction, this disadvantages models by limiting their available context. Second, we noted that BrainJEPA adopts batch normalization prior to the linear probing layer. We therefore standardized the evaluation for all models by (1) sampling eight equally-spaced crops at test time and averaging their logits, and (2) adding a batch normalization layer prior the linear probe layer. We found that both factors dramatically improve the linear probing performance of BrainLM (Table 15).

To understand the surprising impact of batch normalization, we performed a follow-up analysis. As such a large effect likely arises from simple yet highly relevant information, we hypothesized it results from the mean BOLD signal per ROI. Specifically, BrainLM's application of robust scaling preserves differences in the mean signal for each ROI, a feature we have found to be highly predictive in internal analyses. To test this, we pretrained a new BrainLM model on data that was instead z-scored per ROI, which sets the mean of every ROI to zero and thus removes this signal. Indeed, we find this model performs poorly with a linear probe, and crucially, the substantial benefit of batch normalization vanishes (Table 2).

We note that we similarly use multiple crops at test time for fine-tuning. Finally, whereas these analyses use only the CLS-token for BrainLM, we concatenate the CLS and average-pooled tokens

for the main analyses. This standardizes the evaluation with Brain-Semantoks and we observe a mild further improvement (86.7%).

BrainLM Linear Probing Performance for UKB Sex Prediction.

Table 15: Effect of batch normalisation (BN) prior to the linear layer and ensembling over multiple crops.

| # Crops | BN | Accuracy (%) |
|---------|-----|--------------|
| 1 Crop | No | 59.6 |
| 1 Crop | Yes | 78.2 |
| 8 Crops | Yes | 86.0 |

Table 16: Ablation on data normalization. † Reproduction by pretraining from scratch using per-roi Z-scoring.

| Normalization | BN | Accuracy (%) |
|---------------|-----|--------------|
| Robust Scaling | Yes | 86.0 |
| Per-ROI Z-scoring† | No | 58.7 |
| Per-ROI Z-scoring† | Yes | 59.3 |

### A.7 TASK-BASED FMRI

We provide an initial extension to task-based fMRI prediction using short sequences. The Hariri emotion task has a blocked design, in which participants need to match either shapes or faces, with five blocks for each condition. We aim to decode which of these two block types is active during a short fMRI segment. Brain-JEPA and Brain-Semantoks are pretrained on long fMRI sequences and we thus construct a single temporal patch from task. To determine which (if any) block is active at a given time point, we use the HRF-convolved design matrix. We consider the following three options for the construction of a single temporal patch, which we visualize in Figure 7:

- **1 Block (Pad)**: We select the time points for a single block by thresholding the condition-specific regressor in the design matrix at 0.1. As this tends to result in roughly ≈20 seconds of data, we zero-pad the data to the required patch length (≈32 and 40 seconds for Brain-JEPA and Brain-Semantoks, respectively).

- **2 Blocks (Cont.)**: As the task design features two consecutive blocks for each condition twice, we sample a continuous crop of fMRI data from such a double-block period with a random starting point. While this foregoes the need to pad, the data includes a fixation/rest period in the middle.

- **2 Blocks (Cat.)**: We sample twice as in the **1 Block (Pad)** method without replacement, and concatenate the two blocks. If the two blocks are too long to fit in the temporal patch, both are cropped equally.

Given a single temporal patch of data, we zero-pad the temporal dimension to obtain the full input shape the model expects. Next, we use the model's learned mask token to replace the zero-padded data. As a mild form of data augmentation, we assign the patch of real data as either the first or last patch. This is in-distribution for both models, as both are pretrained with similar temporal masks.

Finally, we note that compared to the resting-state pretraining data, the bask-based data is preprocessed differently. Specifically, it includes a 5mm spatial filter and lacks the application of ICA+FIX for confound removal. This, in addition to the presence of task-evoked activity, constitutes a domain-gap despite also being UK-Biobank data.

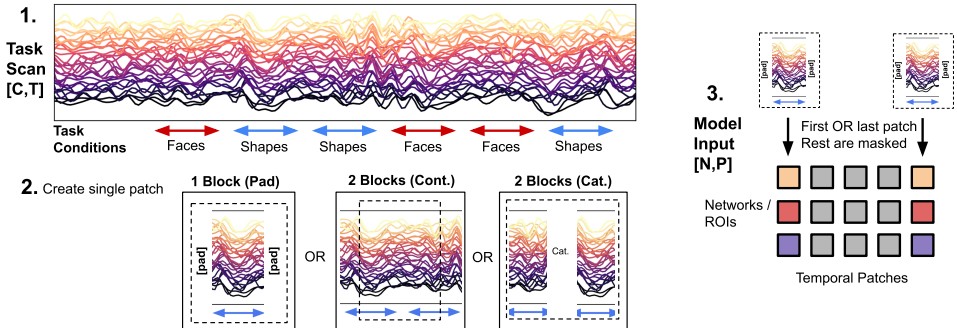

Figure 7: Task-based fMRI data preparation.

## A.8 ADDITIONAL VISUALISATIONS

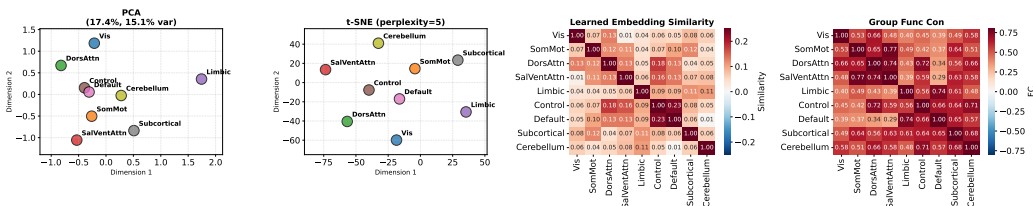

Figure 8: Visualisations of learned network embeddings.

**Learned Network Embeddings.** We inspect the learned positional embeddings for each of the nine networks following pretraining (Figure 8). We summarize our main observations below:

- The magnitude of similarity values between network embeddings are modest, indicating each network is represented distinctly.
- Task-positive network cluster: The model identifies the well-established cluster between the Control network, the Dorsal Attention network, and the Salience/Ventral Attention network.
- Sensory network segregation: The Visual network is segregated from most higher-order association networks.
- Motor, subcortical, limbic pathways: Somatomotor ↔ Subcortical, Somatomotor ↔ Dorsal Attention and Salience/Ventral Attention, Limbic ↔ Subcortical and Cerebellum.
- Control-Default similarity: Surprisingly, the single strongest relationship is between the Control and Default Mode networks. Canonically, these networks are often anti-correlated given their roles in external versus internal processing. However, embedding similarity does not necessitate simple co-activation. The Control network functions as a flexible hub that modulates both the Default Mode Network (DMN; internal) and the Dorsal Attention Network (DAN; external). The high similarities of Control-Default and Control-DorsAttn position the Control network's embedding between these two major cognitive systems, reflecting its known functional topology.

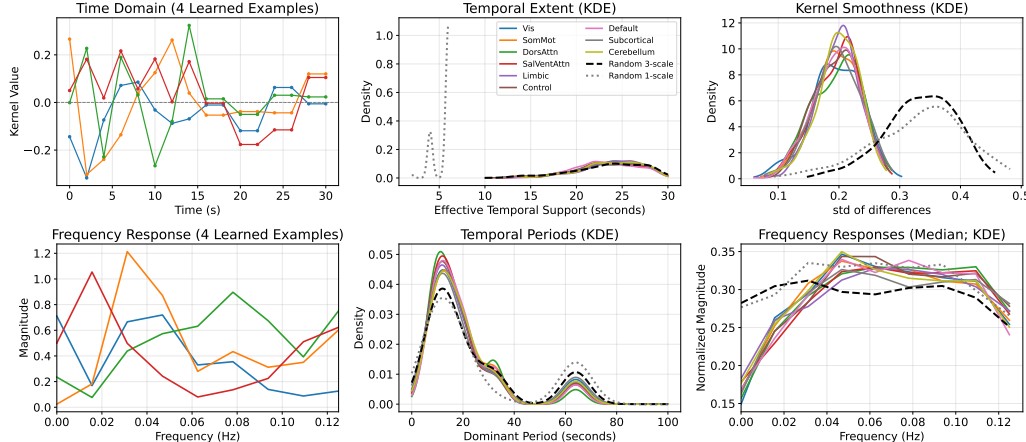

Figure 9: Learned Kernels. Left: Example kernels and their frequency responses. Middle and right: we compute kernel statistics for all $H \times C$ learned kernels and plot the estimated density (KDE). Temporal Extent: We count TRs (2 seconds each) until 90% energy is reached; can be interpreted as the integration window. Kernel smoothness: We compute the standard deviation over diff(kernel). Temporal Periods: We estimate the preferred oscillation timescale by computing $1/\text{argmax(fft)}$. Frequency responses: The median magnitude across frequencies.

## A.9 DETAILS ON HYPERPARAMETERS

Table 17: Hyperparameter settings for all experimental stages.

### (a) Pre-training

| Hyperparameter | Value |
|---|---|
| Optimizer | AdamW |
| Base LR | 0.0007 |
| Epochs | 100 |
| Patch Size | 20 |
| Crop Length | 100 |
| Teacher Momentum | 0.99 |
| Weight Decay | $0.05 \rightarrow 0.3$ |
| Batch Size | 512 |
| Warmup Ratio | 3% (Linear) |
| LR Schedule | Cosine Decay |
| Layer Scale Init | 0.1 |

### (b) Fine-tuning

| Hyperparameter | Value |
|---|---|
| Head Type | Linear Layer |
| Optimizer | AdamW |
| Base LR | 0.0001 |
| Epochs | 50 |
| LR Schedule | Cosine Decay |
| Warmup | None |
| Batch Size | 16 |
| Weight Decay | 0.05 |
| LR Decay Rate | 0.9 |

### (c) Linear Probing

| Hyperparameter | Value |
|---|---|
| Head Type | BN + Linear |
| Optimizer | SGD |
| Momentum | 0.9 |
| Learning Rate | Fixed (best sel.)[a] |
| LR Schedule | None |
| Epochs | 50 |
| Batch Size | $\min(256, n/8)$ |

[a] We fit a linear layer for each of the following learning rates in parallel and choose the best one based on the validation data for test set evaluation: {0.03, 0.01, 0.003, 0.001, 0.0003, 0.0001}

## A.10 STATISTICAL TESTS

Table 18: Comparison of mean balanced accuracy (%) for fMRI time series foundation models using linear probes. Values shown as mean $\pm$ std / p-value (Holm-Bonferroni corrected, Wilcoxon signed-rank test). Best results in **bold**, second-best underlined. **Bold** p-values indicate statistical significance ($p < 0.05$). – indicates best model.

| Model | ABIDE | HBN CELF | HBN WISC | HBN Age | HBN Sex |
|---|---|---|---|---|---|
| BrainLM | 53.84 $\pm$ 3.00 / **0.004** | 42.03 $\pm$ 3.41 / 1.000 | 38.26 $\pm$ 4.11 / 0.160 | 43.89 $\pm$ 2.12 / – | 65.44 $\pm$ 1.72 / **0.004** |
| Brain-JEPA | 52.92 $\pm$ 3.53 / **0.004** | 41.50 $\pm$ 5.16 / 1.000 | 38.34 $\pm$ 3.42 / 0.074 | 39.81 $\pm$ 2.08 / **0.004** | 63.96 $\pm$ 1.45 / **0.004** |
| Brain-Semantoks | **65.13** $\pm$ 2.14 / – | **42.18** $\pm$ 2.80 / – | **40.87** $\pm$ 2.43 / – | 39.16 $\pm$ 0.81 / **0.004** | **69.52** $\pm$ 0.93 / – |

| Model | UKB Age | UKB Sex | SRPBS MDD | SRPBS SZ |
|---|---|---|---|---|
| BrainLM | 30.16 $\pm$ 1.41 / 0.465 | 86.71 $\pm$ 0.63 / **0.027** | 57.61 $\pm$ 4.14 / **0.027** | 55.72 $\pm$ 6.62 / **0.004** |
| Brain-JEPA | 30.60 $\pm$ 2.12 / 0.557 | 83.23 $\pm$ 1.26 / **0.004** | 52.72 $\pm$ 4.18 / **0.004** | 57.63 $\pm$ 3.75 / **0.004** |
| Brain-Semantoks | **31.15** $\pm$ 1.15 / – | **87.52** $\pm$ 0.52 / – | **62.60** $\pm$ 4.79 / – | **69.26** $\pm$ 3.98 / – |

Table 19: Comparison of mean balanced accuracy (%) against supervised and finetuned baselines. Values shown as mean $\pm$ std / p-value (Holm-Bonferroni corrected, Wilcoxon signed-rank test). Best results in **bold**, second-best underlined. **Bold** p-values indicate statistical significance ($p < 0.05$). – indicates best model.

| Model | UKB Age | UKB Sex | HBN Age | HBN Sex | HBN CELF | HBN WISC |
|---|---|---|---|---|---|---|
| FC | 27.04 $\pm$ 1.51 / **0.014** | 80.63 $\pm$ 0.89 / **0.014** | 41.81 $\pm$ 1.36 / **0.014** | 66.51 $\pm$ 1.42 / **0.014** | 42.41 $\pm$ 2.91 / 1.000 | 39.79 $\pm$ 2.91 / 0.826 |
| BNT | 20.48 $\pm$ 1.08 / **0.014** | 77.91 $\pm$ 3.37 / **0.014** | 22.59 $\pm$ 4.31 / **0.014** | 61.74 $\pm$ 9.69 / **0.014** | 42.40 $\pm$ 3.98 / 1.000 | 38.53 $\pm$ 2.94 / 0.523 |
| BolT | 26.85 $\pm$ 1.59 / **0.014** | 80.30 $\pm$ 0.91 / **0.014** | 37.67 $\pm$ 1.41 / **0.014** | 65.22 $\pm$ 1.23 / **0.014** | 42.45 $\pm$ 1.78 / 1.000 | 39.53 $\pm$ 4.42 / 0.826 |
| BrainMass | 23.51 $\pm$ 1.69 / **0.014** | 69.72 $\pm$ 1.89 / **0.014** | 31.80 $\pm$ 1.01 / **0.014** | 56.97 $\pm$ 1.08 / **0.014** | 36.93 $\pm$ 2.06 / **0.014** | 38.00 $\pm$ 2.83 / **0.014** |
| BrainLM | 30.26 $\pm$ 1.66 / **0.014** | 85.75 $\pm$ 0.77 / **0.014** | 39.31 $\pm$ 2.02 / **0.027** | 64.37 $\pm$ 2.32 / **0.014** | 39.27 $\pm$ 4.46 / 0.420 | 35.34 $\pm$ 3.61 / 0.068 |
| Brain-JEPA | 30.60 $\pm$ 1.60 / **0.014** | 86.70 $\pm$ 1.20 / **0.014** | **41.91** $\pm$ 2.00 / – | 65.57 $\pm$ 2.28 / **0.014** | 39.60 $\pm$ 3.50 / **0.014** | 35.20 $\pm$ 3.10 / **0.014** |
| Brain-Semantoks | 31.15 $\pm$ 1.15 / **0.014** | **87.52** $\pm$ 0.52 / – | 39.16 $\pm$ 0.81 / **0.014** | **69.52** $\pm$ 0.93 / – | 42.18 $\pm$ 2.80 / 1.000 | **40.87** $\pm$ 2.43 / – |
| + Finetune | **33.91** $\pm$ 0.87 / – | 87.13 $\pm$ 0.57 / 0.432 | 39.41 $\pm$ 3.77 / 0.084 | 69.31 $\pm$ 0.71 / 0.625 | **42.59** $\pm$ 1.34 / – | 40.82 $\pm$ 1.51 / 1.000 |

| Model | LEMON CVLT | LEMON MDBF | LEMON TMT | ABIDE | SRPBS MDD | SRPBS SZ | Average |
|---|---|---|---|---|---|---|---|
| FC | 39.49 $\pm$ 8.32 / 0.211 | 32.29 $\pm$ 6.09 / 0.195 | 41.14 $\pm$ 6.58 / 0.984 | 65.12 $\pm$ 2.98 / 1.000 | 60.30 $\pm$ 4.65 / 0.211 | **71.59** $\pm$ 5.84 / – | 50.68 / **0.014** |
| BNT | 36.76 $\pm$ 4.90 / **0.020** | 37.90 $\pm$ 5.12 / 0.697 | 33.86 $\pm$ 6.13 / **0.023** | 58.38 $\pm$ 6.51 / 0.109 | 57.60 $\pm$ 4.57 / **0.016** | 66.59 $\pm$ 5.09 / **0.039** | 46.23 / **0.014** |
| BolT | 39.54 $\pm$ 4.71 / 0.111 | 37.98 $\pm$ 5.82 / 0.750 | 40.30 $\pm$ 7.52 / 0.984 | 64.89 $\pm$ 4.08 / 1.000 | 59.50 $\pm$ 4.52 / **0.029** | 67.12 $\pm$ 6.23 / 0.246 | 50.11 / **0.014** |
| BrainMass | 32.24 $\pm$ 3.10 / **0.014** | 31.96 $\pm$ 5.21 / **0.014** | 39.14 $\pm$ 7.00 / **0.014** | 60.89 $\pm$ 3.68 / **0.014** | 59.82 $\pm$ 4.45 / **0.014** | 70.22 $\pm$ 6.20 / **0.014** | 48.43 / **0.014** |
| BrainLM | 37.81 $\pm$ 6.91 / 0.078 | 34.05 $\pm$ 1.84 / 0.068 | 35.33 $\pm$ 3.78 / **0.014** | 53.91 $\pm$ 2.23 / **0.014** | 54.29 $\pm$ 2.38 / **0.014** | 60.10 $\pm$ 5.79 / **0.014** | 47.48 / **0.014** |
| Brain-JEPA | 30.94 $\pm$ 6.20 / **0.014** | 32.26 $\pm$ 6.58 / **0.014** | 35.48 $\pm$ 9.58 / **0.014** | 52.20 $\pm$ 4.00 / **0.014** | 54.00 $\pm$ 4.00 / **0.014** | 60.50 $\pm$ 4.40 / **0.014** | 47.08 / **0.014** |
| Brain-Semantoks | 42.10 $\pm$ 4.72 / 0.375 | **40.23** $\pm$ 5.74 / – | **42.88** $\pm$ 4.05 / – | **65.13** $\pm$ 2.14 / 1.000 | **62.60** $\pm$ 4.79 / 0.652 | 69.26 $\pm$ 3.98 / 0.750 | 52.72 / 0.695 |
| + Finetune | **44.36** $\pm$ 3.36 / – | 38.58 $\pm$ 4.65 / 0.750 | 39.21 $\pm$ 1.91 / 0.082 | **65.44** $\pm$ 1.16 / – | **63.60** $\pm$ 2.56 / – | 71.05 $\pm$ 4.39 / 0.846 | **52.95** / – |

## A.11 MODEL DEVELOPMENT DATA

Model development including extensive evaluations of pretraining stability was carried out using a combination of the UKB dataset (non-overlapping with the holdout set used for our main analyses), a subset of HBN (non-overlapping with samples used for our main analyses), and a large private dataset of resting-state fMRI. The development datasets allowed us to evaluate model transfer and architectural choices using large sample sizes prior to the experiments reported in the main paper.

