# OpenReview forum: "Brain-Semantoks: Learning Semantic Tokens of Brain Dynamics with a Self-Distilled Foundation Model"
_ICLR.cc/2026/Conference — ICLR 2026 Poster_

### Official Review · Reviewer_TTS1 · 2025-10-28

**Soundness:** 3
**Presentation:** 3
**Contribution:** 4
**Rating:** 8
**Confidence:** 4

**Summary:**

This paper presents Brain-Semantoks, a novel fMRI foundation model that shifts the paradigm from reconstruction-based learning to abstraction-based representation learning. The authors propose a neuroscientifically grounded semantic tokenizer that aggregates voxel signals into functionally meaningful network-level tokens, coupled with a self-distillation objective across temporal views to learn stable, high-level representations of brain dynamics. To mitigate instability from noisy fMRI data, a Teacher-Guided Temporal Regularizer (TTR) is introduced to gradually guide the model toward robust convergence. Extensive experiments on multiple public datasets show that Brain-Semantoks achieves state-of-the-art linear probing performance, outperforming both supervised and existing self-supervised methods, with strong generalization across tasks without domain adaptation. Overall, the work makes a compelling case for semantically informed, abstract representation learning in neuroimaging foundation models.

**Strengths:**

1.  Reorients fMRI foundation modeling from masked reconstruction to semantic abstraction via a functional-network tokenizer, aligning tokenization with neuroscientific priors rather than voxel/ROI grain. The self-distillation across temporal crops plus TTR curriculum is a thoughtful adaptation of DINO/iBOT-style ideas to low-SNR fMRI.

2. Solid ablation suite quantifies the effect of tokenizer design (structured conv, patch length), masking scheme/ratio, and TTR duration; results cohere with the hypotheses (e.g., higher masking ratios; network/slice masking reducing interpolation; moderate mask-loss weight).

3. Clear articulation of the problem with reconstruction-centric objectives for phenotype prediction and a well-motivated methodology figure/flow (student-teacher; temporal views; semantic tokens). Related work positioning is balanced.

4. Demonstrates SOTA linear-probe performance across diverse tasks, often surpassing supervised methods, suggesting practically valuable, broadly transferable embeddings; claims of scaling and OOD gains are timely for neuro-foundation models.

**Weaknesses:**

1. Heavy emphasis on linear probing; fewer results on full/partial fine-tuning (or lightweight adapters) and limited analysis of task-based fMRI limits claims about universal transfer beyond resting-state embeddings. The authors acknowledge the task-fMRI gap as future work. Including even a small task-fMRI probe or adapter-tuning study would strengthen external validity.

2. Tokenizer depends on fixed functional networks; while neuro-plausible, robustness to alternative parcellations or data-driven grouping is only proposed as future work. A sensitivity analysis across multiple parcellations (e.g., Schaefer/Glasser granularity) would bolster generality.

3. Cross-dataset variability (hardware/protocols/cohorts) is central; yet the paper could provide a deeper distribution-shift audit (e.g., per-site/per-scanner breakdowns, ComBat-style harmonization baselines, robustness to TR differences) to substantiate OOD claims. Authors note investigating harmful shifts as future work.

4. While ablations are thorough, a targeted component attribution (e.g., semantic tokenizer vs. self-distillation vs. TTR via controlled swaps against strong masked-JEPA baselines) would clarify which element drives most of the gain. Current tables hint at trends but don’t isolate effect sizes against matched alternatives.

5. The “rigorous linear-probe protocol” could use a compact table specifying classifier type, search space, train/val splits, number of repeats, and CIs across all tasks to help reproducibility reviewers quickly verify comparability. (Some of this is in appendices; mirroring a summary in the main text would help.)

**Questions:**

1. Can you report adapter-based or LoRA-style finetuning on a subset of downstream tasks to test whether the learned features remain robust under end-to-end adaptation and to compare against masked-JEPA/BrainLM finetuned baselines?

2. How do results vary across alternative parcellations and numbers of networks/tokens (e.g., Schaefer 100/400, Glasser 360)? You ablate 1/9/20/58 tokens—could you include cross-parcellation comparisons and per-task preferences?

3. Please provide per-site/scanner/TR robustness analyses and/or harmonization baselines (e.g., ComBat) to substantiate OOD claims; which specific shifts most degrade transfer, and does the semantic tokenizer mitigate them?

4. Could you include a matched JEPA/MAE-style latent reconstruction baseline with the same tokenizer and encoder to isolate the benefit of self-distillation + TTR?

5.  Beyond cropping two long segments, did you test time-warping/jittering or frequency-domain perturbations that preserve physiological plausibility? How sensitive is performance to Tcrop?

6. Any early evidence on task-fMRI or clinical endpoints (diagnosis/prognosis) to complement resting-state tasks, even if small-scale?

---

> ### Author Response · Authors · 2025-11-22
> **Rebuttal Reply 1**
>
> We sincerely thank the reviewer for their thorough evaluation of our manuscript as well as their questions, which has enabled us to significantly improve our work. We were especially pleased that the reviewer considered our methodology thoughtful and well-motivated and appreciated the performance improvements and ablation analyses.
>
> We briefly summarize the major extensions and clarifications of the revision, before addressing the questions one-by-one.
>
> - **1. Extension to task-based fMRI**: We show strong transfer of learned representations for short fMRI crops of the Hariri emotion task (l.418 and Table 3, p9).
> - **2. Expanded evaluations**: Additional baseline comparison with BrainMass model (Table 2, p8) and extensive ablations on brain atlases (Section A.4.1, p16), temporal resolution (Section A.4.2, p.16), data augmentations (Section A.4.3, p.16), baseline foundation models (Section A.6, p.18).
> - **3. Additional visualisations** of learned network embeddings (Section A.8, p.20), learned kernel properties (Section A.8, p.21), and OOD generalisation (Section A.5, p.17).
>
> Corrections and clarifications:
> - Preprocessing (Filtering): We have updated the manuscript to explicitly state that inputs are bandpass filtered (0.01-0.1Hz). This standard preprocessing step isolates hemodynamic responses and was utilized in our original experiments but not detailed in the initial text.
> - MDBF: We have corrected our description of the MDBF scale as a mood (state) measure rather than personality (trait) measure, and now provide detailed scale descriptions in section A.2.
>
> ---
>
> **Q1. Can you report adapter-based or LoRA-style finetuning on a subset of downstream tasks to test whether the learned features remain robust under end-to-end adaptation and to compare against masked-JEPA/BrainLM finetuned baselines?**
>
> We thank the reviewer for this question. We would like to clarify that we do include end-to-end finetuning results in Table 2, which directly tests whether learned features remain robust under full adaptation. These results demonstrate that generally our representations maintain their quality when all parameters are updated. Notably, we observe that finetuning baselines (Brain-JEPA, BrainLM) under identical conditions underperform our linear probing results on 11 out of 12 tasks, further validating the quality of our pretrained representations.
>
> Regarding parameter-efficient finetuning methods (adapters, LoRA): as our foundation model is relatively compact (~63M parameters, for reference BrainLM's largest variant is 650M), full finetuning is computationally feasible and we observe stable training. The fact that we see only a minor improvement from finetuning over probing (rather than more significant gains; +0.23% on average) suggests our representations already capture relevant information effectively and thereby reduce the need for extensive adaptation.
>
> That said, if the reviewer is specifically interested in seeing adapter-based or LoRA-style results to compare parameter efficiency or adaptation dynamics, we would be happy to provide these experiments. Please let us know if this would strengthen our evaluation in your view.
>
> ---
>
> **Q2. How do results vary across alternative parcellations and numbers of networks/tokens (e.g., Schaefer 100/400, Glasser 360)? You ablate 1/9/20/58 tokens—could you include cross-parcellation comparisons and per-task preferences?**
>
> We thank the reviewer for their suggestion. We are pleased to present the results of multiple experiments, in which we investigate comparisons to our default atlas combination.
>
> **Default Configuration**
> Schaefer-400 ('S400'), Tian-3, Buckner-7: **52.39%** (Avg Accuracy)
>
> - **Low- and high-resolution cortical parcellations**
>   - S400 → S100: **49.82%**
>   - S400 → S800: **51.06%**
>
> - **Alternative cortical parcellation (Gordon-333, different network aggregation)**
>   - S400 → Gordon-333: **51.48%**
>
> - **Low-resolution subcortical atlas**
>   - Tian-3 (50 ROIs) → Tian-1 (16 ROIs): **51.27%**
>
> - **Higher-resolution cerebellum atlas**
>   - Buckner-7 → Buckner-17: **52.26%**
>
> We present the complete per-dataset results in Section A.4.1 (p16).
>
> Overall, the results are in line with the literature in that a moderate amount of ROIs generally works well. If too few are used, the parcellation step destroys too much signal. Too many and one is liable to model too much noise. We note that for the Gordon-333 atlas [1] we map the 333 cortical ROIs to 13 'communities' using the provided membership; the parcellation thereby offers an alternative aggregation solution to the Yeo-networks. Interestingly, the Gordon-333 solution with a total of 15 spatial tokens still yields strong downstream performance, indicating robustness to the specific network mapping that is chosen.
>
> [1]. Gordon et al., Generation and evaluation of a cortical area parcellation from resting-state correlations. Cerebral cortex. 2016 Jan 1;26(1):288-303.

---

> ### Author Response · Authors · 2025-11-22
> **Rebuttal Reply 2**
>
> **Q3. Please provide per-site/scanner/TR robustness analyses and/or harmonization baselines (e.g., ComBat) to substantiate OOD claims; which specific shifts most degrade transfer, and does the semantic tokenizer mitigate them?**
>
> We thank the reviewer for this interesting suggestion. We are pleased to present additional robustness analyses pertaining to out-of-distribution generalization. Specifically, we create data subsets based on site, MRI scanner, repetition time (TR), spatial resolution, and field-of-view (FOV). For both SRPBS and ABIDE we perform sex and age prediction to maximize sample size availability. We probe Brain-Semantoks, as well as versions trained without the global CLS loss or without the Semantic Tokenizer (i.e., using ROI-wise linear projection). We include these new analyses with visualisations in Section A.5 (p17).
>
> For SRPBS, the two ablated models exhibit stronger performance drops for increased TR, lowered resolution, and lowered FOV. Yet the overall results indicate that both components are important for consistent performance. However, it is not obvious from these results that certain investigated scanning parameters are particularly problematic for transfer learning. Although these analyses are not exhaustive and therefore do not preclude the possibility that other factors would provide critical insight, these findings suggest uniform robustness improvements from the complete model. Such patterns may result from generally improved representation quality that is robust to scanning parameter variations rather than mere overfitting to pretraining data characteristics.
>
> ---
>
> **Q4. Could you include a matched JEPA/MAE-style latent reconstruction baseline with the same tokenizer and encoder to isolate the benefit of self-distillation + TTR?**
>
> We thank the reviewer for highlighting a lack of clarity in our paper. This is included in our ablations, but we recognize it is not appropriately structured to clearly convey this. The contributions of the tokenizer (Figure 4), self-distillation of CLS loss, and TTR are evaluated separately.
>
> Regarding the impacts of self-distillation and TTR impacts, we compare to our full model with 52.39\% average downstream accuracy:
>
> **TTR**: Table 5 (p10): Omitting TTR (i.e. a duration of 0) achieves 50.88%
>
> **CLS Loss**: Table 7 (p10), row 1 (CLS=No, $\lambda_{Tok}$=1.0) represents a JEPA-style token reconstruction-only approach using our semantic tokenizer and encoder, achieving 47.32\%
>
> We have now updated the table caption to clarify this.
>
> ---
>
> **Q5 .Beyond cropping two long segments, did you test time-warping/jittering or frequency-domain perturbations that preserve physiological plausibility? How sensitive is performance to Tcrop?**
>
> We thank the reviewer for their question. We are pleased to provide additional ablation experiments on data augmentations, where we focus on physiological plausibility and frequency-based perturbations.
>
> Specifically, we tested:
> - **ROI Mixing:** We mix signals between ROIs based on spatial adjacency to mimic partial voluming, registration errors, or head motion.
> - **Frequency Masking:** Dampens power in certain parts of the frequency spectrum (0.01-0.1Hz).
> - **Band-Specific Noise:** Perturbs signal in a frequency band by mixing in sinusoidal noise.
>
> We observe the following average performance across downstream tasks:
>
> - Temporal views only: 51.91% accuracy
> - \+ Physiological augs (light): 51.85%
> - \+ Physiological augs (heavy): 51.92%
> - \+ Corruption augs (light, as in paper): **52.39%**
>
> Surprisingly, physiologically-motivated augmentations provide no benefit over temporal view selection alone, while simple corruption augmentations yield a small improvement. We hypothesize that (1) fMRI data already contains substantial physiological noise, so our cross-view distillation objective implicitly learns invariance to these factors without explicit augmentation. (2) Our input data follows a conventional bandpass filter (0.01-0.1Hz) to isolate hemodynamic responses. Consequently, we were constrained to applying frequency-based augmentations within this signal-rich band. This likely creates a destructive trade-off: attempting to simulate spectral noise in this range and introduce data diversity risks corrupting the underlying neural signal. Similarly, we excluded global spike augmentations (as an alternative to mimic head motion effects) as such artifacts are inconsistent with bandpass-filtered inputs.
>
> (Reply continued in Rebuttal Reply 3)

---

> ### Author Response · Authors · 2025-11-22
> **Rebuttal Reply 3**
>
> (Continued answer to Q5)
>
> The effectiveness of corruption-based augmentations may stem from their ability to increase data diversity without introducing structured artifacts that could interfere with the semantic-level representations our tokenizer targets. However, we acknowledge this augmentation exploration is limited, and more sophisticated physiological noise models remain an interesting direction for future work. We have added these results to Section A.4.3 (p16).
>
> We also explored time-warping early in development but found it degraded performance fairly significantly. We suspected this might result from the augmentations violating the temporal smoothness characteristic of the BOLD signal.
>
> On the sensitivity to Tcrop: We are currently running ablations for this. Unfortunately, due to the large amount of total experiments, some of our compute jobs have been stuck in queue. We will add a comment with results as soon as they complete and apologize for the delay.
>
> Edit: We have now completed the experiment on Tcrop and describe the results in a comment below.
>
> ---
>
> **Q6. Any early evidence on task-fMRI or clinical endpoints (diagnosis/prognosis) to complement resting-state tasks, even if small-scale?**
>
> We appreciate this question. Our paper already includes the following clinical diagnostic tasks: schizophrenia, autism spectrum disorder, and major depressive disorder (Table 1 and 2). Their results demonstrate that Brain-Semantoks learns clinically-relevant representations from resting-state data.
>
> Additionally, we are pleased to present an extension of our evaluation to task-based fMRI. For this, we use the Hariri emotion task from UK Biobank, where participants match either shapes or emotional faces in a blocked design. We formulate this as predicting block type ('shapes' vs. 'faces') from short fMRI segments.
>
> This constitutes a challenging transfer despite using UKB data for pretraining. First, task-based fMRI introduces a substantial domain gap: it uses different preprocessing and contains task-evoked activity absent from resting-state data. Second, the task presents a fundamentally different prediction problem, as it requires discriminating within-subject temporal dynamics rather than between-subject phenotypes. Third, task blocks are substantially shorter than pretraining sequences, requiring the model to generate summary representations from limited context.
>
> We address the temporal mismatch by leveraging our masked distillation framework: we construct a single temporal patch from a task block and mask all remaining positions, matching the objective optimized during pretraining. We test three strategies for patch creation (zero-padding, concatenating contiguous blocks, or concatenating timeseries from separate blocks; see Section A.7 (p19) for methodological details).
>
> Table 3 (copied below) shows Brain-Semantoks substantially outperforms Brain-JEPA across all settings, achieving >12% improvement with linear probing} and maintaining strong advantages with finetuning. This demonstrates effective transfer to a different fMRI modality, task paradigm, and temporal scale. We now describe these results in the paper starting at L.417 (p8).
>
>
> | **Model**              | **1 Block**    | **2 Blocks (Cont.)** | **2 Blocks (Cat.)** |
> |------------------------|----------------|----------------------|---------------------|
> | **_Linear Probing_**  |                |                      |                     |
> | Brain-JEPA            | 81.45 ± 0.59  | 82.29 ± 0.28        | 81.06 ± 0.81       |
> | Brain-Semantoks       | **93.84 ± 0.36** | **94.34 ± 0.75**    | **96.50 ± 0.15**   |
> | **_Finetuning_**      |                |                      |                     |
> | Brain-JEPA            | 91.04 ± 1.56  | 92.33 ± 1.86        | 94.71 ± 0.85       |
> | Brain-Semantoks       | **96.89 ± 0.73** | **97.85 ± 0.86**    | **97.70 ± 0.80**   |
>
> ---
>
> Finally, we thank the reviewer again for their help in evaluating our work.

---

> ### Author Response · Authors · 2025-11-23
> **Rebuttal Reply 4: Tcrop ablation**
>
> **How sensitive is performance to Tcrop?**
>
> To address this we perform a post-hoc analysis on the sensitivity to $T_{\text{crop}}$ by pretraining with crop lengths of 80, 100, and 120 timepoints and measuring average performance across our 10 downstream tasks.
>
> | $T_{crop}$ | Multi-crop (8×) | Single crop |
> |-------|----------|----------|
> | 80    | 51.70    | 50.58 |
> | 100   | 52.39    | 51.33 |
> | 120   | 52.24    | 51.60 |
>
> Our default setup averages prediction logits across 8 crops per sample. We observe that performance does not increase monotonically with crop length. Longer crops provide more information per sample and indeed single-crop prediction does improve monotonically (50.58% → 51.33% → 51.60%). However, we hypothesize this is offset by several factors:
>
> - Distillation task difficulty: Longer crops make the student-teacher matching task easier, reducing pressure to learn strong, generalizable representations.
> - Data diversity: Longer crops reduce the number of distinct crops per recording, decreasing sample-level diversity during pretraining.
> - Ensembling benefits: Longer crops have greater overlap, reducing the benefit of multi-crop ensembling during inference.
>
> Overall, the method appears reasonably robust across this range.
>
> We have added the ablation to Section A.12 (p22).

---

### Official Review · Reviewer_sTj7 · 2025-10-31

**Soundness:** 3
**Presentation:** 3
**Contribution:** 2
**Rating:** 4
**Confidence:** 4

**Summary:**

The paper proposes Brain‑Semantoks, a self‑supervised fMRI foundation model that shifts from reconstruction to abstraction by (1) introducing a semantic tokenizer that aggregates ROI time series into network‑level tokens and (2) pretraining via self‑distillation across long temporal views with a Teacher‑guided Temporal Regularizer (TTR) for stability on noisy fMRI signals. The model shows strong linear‑probe performance across nine downstream datasets/tasks (ABIDE, HBN, UKB, SRPBS), outperforming BrainLM and Brain‑JEPA and often rivaling or surpassing fully supervised/fine‑tuned baselines, with ablations and scaling analyses.

**Strengths:**

- Clear reframing toward semantic abstraction with a neuroscience‑grounded tokenizer operating at functional network granularity, reducing token length and noise while injecting inductive bias.

- The slice masking to avoid trivial interpolation is a strong regularization that forces the model to learn meaningful relationships between tokens.

- Well‑designed curriculum via TTR that averages network tokens over time early in training, improving stability of the model during training

- Rigorous linear‑probe protocol with 10‑fold CV and standardized heads; strong results across clinical and demographic tasks

- Detailed scaling analysis for fMRI FMs under linear probing, showing power‑law like gains with pretraining size and OOD improvements without domain adaptation.

**Weaknesses:**

- The atlas choice is mostly arbitrary. No analysis of how results change with alternative parcellations (Schaefer, Shen, Yeo‑17) or different subcortical/cerebellar groupings; no exploration of data‑driven network discovery to justify the choice of nine functional networks.

- The geometry of learned network identity embeddings is not analyzed; it is unclear whether they capture canonical inter‑network relationships or known hierarchies.

- Precise kernel sequences and decay parameters are unclear; there is no connection to hemodynamic time constants nor visualization of learned filters or frequency responses to support the inductive bias.

- The scaling of the coding‑rate regularizer with batch size and feature dimension is not described; the influence of batch‑dependent normalization on the covariance and the regularizer is not evaluated.

- There is no evaluation of physiologically realistic, band‑limited, or acquisition‑mimicking perturbations (e.g., motion, respiratory/cardiac noise, TR/resampling artifacts) beyond simple channel/time zeroing, Gaussian noise, and amplitude scaling.

**Questions:**

- Functional networks: Reducing the complexity of the input data via the use of an atlas for grouping based on functional network annotation is a sensible choice. However, as briefly discussed by the authors, there are several alternatives for such atlases (eg, Schaefer, Shen or even Yeo-17 [see ref1 for other atlases]), from which Yeo-7, in addition to the two subcortical networks, is the most minimalist. How sensitive are the results to the chosen atlas and the definition of subcortical/cerebellar groupings? Have you evaluated alternative parcellations? Have you considered a data‑driven network discovery for an appropriate number of functional networks?

- Positional and network embeddings: Does the model learn network identity embeddings that capture known functional relationships? Any probing of these embeddings’ geometry vs canonical network hierarchies?

- Structured convolutions: What precise kernel sequences and decay profiles were used, and how do they relate to expected hemodynamic time constants? Can you share learned filters or frequency responses to justify the inductive bias?

- Coding‑rate regularizer: how is R scaled with batch size and feature dimension; what is the effect of batch‑dependent normalization on R?

- Augmentations: The paper uses light corruptions (channel/time zeroing, Gaussian noise, amplitude scaling), which likely result in out-of-distribution data. Have you tested physiological noise models or band‑limited perturbations that mimic TR/resampling artifacts?

References:

[ref1] https://www.lead-dbs.org/helpsupport/knowledge-base/atlasesresources/cortical-atlas-parcellations-mni-space/

---

> ### Author Response · Authors · 2025-11-22
> **Rebuttal Reply 1**
>
> We sincerely thank the reviewer for their thorough evaluation of our manuscript as well as their questions, which has enabled us to significantly improve our work. We were especially pleased that the reviewer appreciated multiple of our model design choices as well as the strong results from extensive downstream performance and scaling evaluations.
>
> We briefly summarize the major extensions and clarifications of the revision, before addressing the questions one-by-one.
>
> - **1. Extension to task-based fMRI**: We show strong transfer of learned representations for short fMRI crops of the Hariri emotion task (l.418 and Table 3, p9).
> - **2. Expanded evaluations**: Additional baseline comparison with BrainMass model (Table 2, p8) and extensive ablations on brain atlases (Section A.4.1, p16), temporal resolution (Section A.4.2, p.16), data augmentations (Section A.4.3, p.16), baseline foundation models (Section A.6, p.18).
> - **3. Additional visualisations** of learned network embeddings (Section A.8, p.20), learned kernel properties (Section A.8, p.21), and OOD generalisation (Section A.5, p.17).
>
> Corrections and clarifications:
> - Preprocessing (Filtering): We have updated the manuscript to explicitly state that inputs are bandpass filtered (0.01-0.1Hz). This standard preprocessing step isolates hemodynamic responses and was utilized in our original experiments but not detailed in the initial text.
> - MDBF: We have corrected our description of the MDBF scale as a mood (state) measure rather than personality (trait) measure, and now provide detailed scale descriptions in section A.2.
>
>
> **Q1: (...) How sensitive are the results to the chosen atlas and the definition of subcortical/cerebellar groupings? Have you evaluated alternative parcellations? Have you considered a data‑driven network discovery for an appropriate number of functional networks?**
>
> We thank the reviewer for their question, we agree this aspect was under-discussed in our manuscript. The Schaefer atlases have been found both in the literature and internal analyses to enable good predictive performance. We decided to match to the previous state-of-the-art, Brain-JEPA, by using Schaefer-400 (S400) and their chosen subcortical Tian-3 atlas. We decided to complement this choice by including an atlas of the cerebellum (Buckner-7), which we believed to be sensible for a foundation model. Still, an empirical evaluation of these choices was lacking. We are pleased to present the results of multiple experiments, in which we compare to our default atlas combination.
>
> **Default Configuration**
> Schaefer-400 ('S400'), Tian-3, Buckner-7: **52.39%** (Avg Accuracy)
>
> - **Low- and high-resolution cortical parcellations**
>   - S400 → S100: **49.82%**
>   - S400 → S800: **51.06%**
>
> - **Alternative cortical parcellation (Gordon-333, different network aggregation)**
>   - S400 → Gordon-333: **51.48%**
>
> - **Low-resolution subcortical atlas**
>   - Tian-3 (50 ROIs) → Tian-1 (16 ROIs): **51.27%**
>
> - **Higher-resolution cerebellum atlas**
>   - Buckner-7 → Buckner-17: **52.26%**
>
> We present the complete per-dataset results in Section A.4.1 (p16), which also includes our interpretation.
>
> Overall, the results are in line with the literature in that a moderate amount of ROIs generally works well. If too few are used, the parcellation step destroys too much signal. Too many and one is liable to model too much noise. We note that for the Gordon-333 atlas [1] we map the 333 cortical ROIs to 13 'communities' using the provided membership; the parcellation thereby offers an alternative aggregation solution to the Yeo-networks. Interestingly, the Gordon-333 solution with a total of 15 spatial tokens still yields strong downstream performance, indicating robustness to the specific network mapping that is chosen.
>
> Finally, regarding data-driven network discovery: yes, we think this is a very interesting research question and we did consider it at the beginning of the project. However, we did not pursue it for the following reasons. First, there exists impressive literature that generated solutions like Yeo-7, Yeo-17, and Gordon communities, with extensive post-hoc evaluations. This implies a fairly high bar to pass, and improvements would likely be non-trivial to achieve. Second, although we hypothesized that ~450 spatial tokens are too many for fMRI transformers, we had yet to establish they could be successfully compressed into as few as ~9. We see the current work as an important validation of this idea, which also allowed for the testing of alternative pretraining strategies, which we deemed important to make progress on.
>
> That being said, we do believe there is promise in learning the mappings from data. First, much methodological progress has been made since Yeo et al., and larger datasets have become available. However, we suspect improvements over Yeo will be relatively incremental compared to the contributions of the current paper.

---

> ### Author Response · Authors · 2025-11-22
> **Rebuttal Reply 2**
>
> **Q2: Positional and network embeddings: Does the model learn network identity embeddings that capture known functional relationships? Any probing of these embeddings’ geometry vs canonical network hierarchies?**
>
> We thank the reviewer for this interesting comment, which prompted us to perform a more detailed analysis of the learned network embeddings. We have now conducted this analysis and included it in the revised manuscript, including dimensionality reduction visualizations (PCA, t-SNE) and an embedding similarity heatmap (Section A.8, p.20)].
>
> We outline the key observations below:
>
> - The magnitude of similarity values between network embeddings are modest, indicating each network is represented distinctly.
> - Task-positive network cluster: The model identifies the well-established cluster between the Control network, the Dorsal Attention network, and the Salience/Ventral Attention network.
> - Sensory network segregation: The Visual network is segregated from most higher-order association networks.
> - Motor, subcortical, limbic pathways: Somatomotor ↔ Subcortical, Somatomotor ↔ Dorsal Attention and Salience/Ventral Attention, Limbic ↔ Subcortical and Cerebellum.
> - Control-Default similarity: Surprisingly, the single strongest relationship is between the Control and Default Mode networks. Canonically, these networks are often anti-correlated given their roles in external versus internal processing. However, embedding similarity does not necessitate simple co-activation. The Control network functions as a flexible hub that modulates both the Default Mode Network (internal processing) and the Dorsal Attention Network (external processing). The high similarities of Control-Default and Control-DorsAttn position the Control network's embedding between these two major cognitive systems, reflecting its known functional topology.
>
> **Q3: Structured convolutions: What precise kernel sequences and decay profiles were used, and how do they relate to expected hemodynamic time constants? Can you share learned filters or frequency responses to justify the inductive bias?**
>
> We thank the reviewer for highlighting the omission of detailing the exact kernel structure that was used. We now include this information in the paper (section A.3, p15) and clarify our argumentation for the chosen kernel structure for BOLD signals:
>
> The Conv-STD  branch uses short kernels for local, cross-ROI dependencies. Meanwhile, the Conv-STR branch uses longer, decaying kernels to enforce a temporal inductive bias absent in standard ViT architectures. Specifically, due to the hemodynamic properties of the BOLD signal, it is both highly autocorrelated and smooth. We therefore hypothesize a temporal locality prior is beneficial, combined with a longer kernel to model the slow-evolving signal.
>
> To transform the input time series from $C_n$ ROIs to $\frac{D}{2}$ channels, an expansion factor is computed as $H = \text{ceil}\left( \frac{D/2}{C_n}\right)$ ('heads'). Each ROI is then processed by its set of $H$ filters, resulting in $H \times C_n$ total channels. We furthermore adopt a learnable residual connection as used in [Li et al., 2022; Vetter et al., 2024], which is analogous to the direct feedthrough term used in state-space models. Finally, a linear projection layer maps from $H \times C_n$ to the target $\frac{D}{2}$ dimensionality, ensuring a fixed output dimensionality while also mixing information across ROIs.
>
> Whereas the standard convolutions (Conv-STD use a kernel size of 3, the structured, depthwise-separable convolutions (Conv-STR) use a base kernel size of 4 across 3 scales. Specifically, a kernel of length 16 is created by concatenating three kernels: a 4-length kernel with weight 4.0, a 4-length kernel with weight 2.0, and an 8-length kernel (upsampled from 4 parameters) with weight 1.0.
>
> ---
>
> We additionally appreciate the insightful request to inspect the learned filters. We have now done so and include these in section A.8 (p21). These visualisations include example kernels and their frequency responses. We furthermore plot distributions over various properties of the learned filters while separating based on which network-tokenizer they came from. We here include our main observations:
> 1. Learned filters are well matched to the BOLD signal and are found to encode slow oscillations.
> 2. Following pretraining, kernels have become much smoother than at initialization.
> 3. The frequency responses are mildly downweighting frequencies under 0.04Hz, but the entire physiological range (e.g. 0.01-0.1Hz) is well covered.
> 4. We found these learned properties (smoothness, band-pass characteristics) were highly consistent, with only minor differences across functional networks. This indicates the model captured a general property of the BOLD signal, rather than an idiosyncratic feature of one network.

---

> ### Author Response · Authors · 2025-11-22
> **Rebuttal Reply 3**
>
> **Q4: Coding‑rate regularizer: how is R scaled with batch size and feature dimension; what is the effect of batch‑dependent normalization on R?**
>
> We thank the reviewer for highlighting that these details were not sufficiently described. Generally we adopted the specific implementation details from the SimDINO codebase (Wu et al., 2025).
>
> On scaling with batch size and feature dimension: Instead of manual tuning or the theoretical gradient-norm scaling mentioned in the original manuscript, we utilized the batch-balancing heuristic provided in the official implementation. The regularization weight $\gamma$ is scaled dynamically as $\gamma = \frac{D+B}{DB}$ where $D$ is the feature dimension and $B$ the effective batch size. The heuristic ensures that the regularisation strength is adjusted for varying batch sizes.
>
> On the regularizer formulation: We also follow SimDINO's formulation of the coding rate. While the theoretical definition often scales $1/\epsilon^2$, we adopt a linear scaling: $R_\epsilon(\Sigma) = \frac{1}{2}\log\det(I + \frac{D}{\epsilon}\Sigma)$, where $\Sigma$ is the batch-estimated covariance and $\epsilon=0.05$.
>
> On batch-dependent estimation: Since $\Sigma$ is estimated on batches, the balancing factor $\gamma$ described functions as a normalizer to counteract sampling variance and batch size scaling effects. We found that in our experiments the heuristic provided stable convergence without needing to tune extra hyperparameters.
>
> We have updated our Methods section accordingly to make our implementation explicit.
>
> **Q5: Augmentations: The paper uses light corruptions (channel/time zeroing, Gaussian noise, amplitude scaling), which likely result in out-of-distribution data. Have you tested physiological noise models or band‑limited perturbations that mimic TR/resampling artifacts?**
>
> We thank the reviewer for this insightful suggestion. Motivated by this, we conducted additional experiments comparing physiologically-grounded augmentations to our corruption-based approach. Specifically, we tested:
>
> - **ROI Mixing:** We mix signals between ROIs based on spatial adjacency to mimic partial voluming, registration errors, or head motion.
> - **Frequency Masking:** Dampens power in certain parts of the frequency spectrum (0.01-0.1Hz).
> - **Band-Specific Noise:** Perturbs signal in a frequency band by mixing in sinusoidal noise.
>
> We observe the following average performance across downstream tasks:
>
> - Temporal views only: 51.91% accuracy
> - \+ Physiological augs (light): 51.85%
> - \+ Physiological augs (heavy): 51.92%
> - \+ Corruption augs (light, as in paper): **52.39%**
>
> Surprisingly, physiologically-motivated augmentations provide no benefit over temporal view selection alone, while simple corruption augmentations yield a small improvement. We hypothesize that (1) fMRI data already contains substantial physiological noise, so our cross-view distillation objective implicitly learns invariance to these factors without explicit augmentation. (2) Our input data follows a conventional bandpass filter (0.01-0.1Hz) to isolate hemodynamic responses. Consequently, we were constrained to applying frequency-based augmentations within this signal-rich band. This likely creates a destructive trade-off: attempting to simulate spectral noise in this range and introduce data diversity risks corrupting the underlying neural signal. Similarly, we excluded global spike augmentations (as an alternative to mimic head motion effects) as such artifacts are inconsistent with bandpass-filtered inputs.
>
>
> The effectiveness of corruption-based augmentations may stem from their ability to increase data diversity without introducing structured artifacts that could interfere with the semantic-level representations our tokenizer targets. However, we acknowledge this augmentation exploration is limited, and more sophisticated physiological noise models remain an interesting direction for future work. We have added these results to Section A.4.3 (p16).
>
> ---
>
> Finally, we would like to say again that we sincerely appreciate the help of the reviewer and their useful evaluation of our work.
>
> [1]. Gordon EM, Laumann TO, Adeyemo B, Huckins JF, Kelley WM, Petersen SE. Generation and evaluation of a cortical area parcellation from resting-state correlations. Cerebral cortex. 2016 Jan 1;26(1):288-303.

---

> ### Comment · Reviewer_sTj7 · 2025-11-27
>
> (Q1) I thank the authors for adding the atlas ablation studies. The results show that atlas choice mainly produces modest shifts in performance and that medium‑resolution, functionally informed parcellations offer the best trade‑off between stability and expressivity, suggesting that the main conclusions do not hinge on a single, arbitrary atlas.
>
> (Q2) I appreciate the additional analysis of the network embeddings. It is reassuring to see that the model is not simply inheriting atlas labels, but instead learns an internal geometry over networks that mirrors known functional hierarchies. This supports the claim that the semantic tokens capture meaningful neurobiological structure rather than arbitrary groupings.
>
> (Q3) Thank you for the detailed description of the convolutional branches. The current results already show that a fully learned long temporal kernel (“Full kernel (20)”) underperforms both the short‑kernel conv and the structured Conv‑STR branch, supporting your claim that a strong temporal inductive bias is beneficial in this noisy fMRI regime. However, my original question remains: to what extent does the observed Conv‑STR behavior (matching BOLD properties) emerge from the data versus being largely dictated by the structured kernel design? The filter visualizations in Appendix A.8 nicely confirm that Conv‑STR behaves as intended (smooth, long‑range, band‑pass filters in the 0.01–0.1 Hz range), but given that Conv‑STR is explicitly constructed from multi‑scale, decaying kernels on band‑passed BOLD data, such properties are at least partly expected. To further disentangle the effect of the structured design from simple capacity, I would encourage one additional control: replace Conv‑STR with a second, fully learned temporal conv branch (e.g., a longer or dilated conv with similar receptive field and parameter count), while retaining the two‑branch architecture (“Short + Full‑kernel”) and training it end‑to‑end. Comparing “Short + Struct” to this “Short + learned long‑conv” baseline would clarify whether the gains come specifically from the structured multi‑scale, decaying kernel family, rather than just from having an extra convolutional path.
>
> (Q4) Thank you as well for the detailed clarification of the coding‑rate implementation and its batch/feature‑dependent scaling; this fully answers the “how” part of my question. I am still unsure, though, about its role and necessity in your setup. Have you tried removing this term entirely? If so, what is the impact on training stability and linear‑probe performance? Showing such a variant would directly demonstrate the stabilizing effect of R and its interaction with batch‑dependent normalization in your specific application, rather than relying only on intuition and prior SimDINO results.
>
> (Q5) I appreciate the additional augmentation experiments. The discussion of the risk of corrupting neural signal when injecting additional in‑band “physiological” noise is helpful, and it addresses my concern about whether more realistic noise models or TR/resampling‑like perturbations had been considered.
>
> Overall, the revisions and additional analyses substantially strengthen the manuscript and address most of my original concerns, particularly regarding atlas sensitivity, network interpretability, convolutional design details, and augmentation strategy. The remaining suggestions above (a two‑branch “Short + learned long‑conv” control and a no‑coding‑rate variant) are not criticisms of the current results, but rather proposals for clarifying how much of the observed behavior is driven by architectural priors versus learned from data, and for further isolating the contribution of the coding‑rate term in this specific application.

---

> > ### Author Response · Authors · 2025-12-02
> >
> > We are very pleased to hear that our additional experiments have substantially strengthened our paper and addressed most of the reviewer's initial concerns. We sincerely thank the reviewer for their continued thoughtful evaluation of our work and their control analysis suggestions.
> >
> > Regarding the control analyses, we agree on both accounts and have run the requested experiments. We summarize the results below:
> >
> > ---
> >
> > Q3. **Two-branch kernel control:** To disentangle the effect of the structured design from model capacity, we evaluated the requested "Short + Learned Long-Conv" baseline, using a carefully matched (unstructured) kernel. Specifically, we replaced the structured kernel branch with a standard, learned depthwise convolutional kernel with the exact same receptive field (16) and a comparable parameter count. The results are compared below:
> >
> > | **Architecture** | **Kernel Configuration**           | **Score** |
> > | ---------------- | ---------------------------------- | --------- |
> > | **Control**      | Short (3) + Learned Long-Conv (16) | 51.35     |
> > | **Ours**         | Short (3) + Structured (16)        | **52.39** |
> >
> > Despite the greater flexibility of the standard convolution, we observe improved performance with the structured kernel. This confirms that the performance gains are driven by the specific temporal inductive bias (decay profile) rather than simply having a second convolutional path with a larger receptive field. We now include this ablation in table 4 (p10).
> >
> > ---
> >
> > Q4. **No-coding-rate variant**: We pretrain without the coding rate regularizer and observed immediate model collapse. Specifically, cosine similarities between student and teacher models of 1 indicate they are identical which prevents learning via self-distillation. This matches the theoretical work in literature such as SimDINO [1] but also prior works such as VICReg [2]. We have included this ablation with training loss visualisations in Appenddix A.13 (p23).
> >
> >
> > [1] Wu, Z., Zhang, J., Pai, D., Wang, X., Singh, C., Yang, J., ... & Ma, Y. (2025). Simplifying dino via coding rate regularization. arXiv preprint arXiv:2502.10385.
> >
> > [2] Bardes, A., Ponce, J., & LeCun, Y. (2021). Vicreg: Variance-invariance-covariance regularization for self-supervised learning. arXiv preprint arXiv:2105.04906.
> >
> > ---
> >
> > Finally, we thank the reviewer again for their work which we believe allowed us to significantly improve our manuscript.

---

### Official Review · Reviewer_C6gR · 2025-10-31

**Soundness:** 3
**Presentation:** 3
**Contribution:** 3
**Rating:** 8
**Confidence:** 4

**Summary:**

This paper introduces Brain-Semantoks, a self-supervised fMRI foundation model that combines a semantic tokenizer and a student–teacher self-distillation framework to learn high-level, temporally stable representations of brain dynamics. Instead of voxel- or region-level reconstruction, the model produces recording-level embeddings based on functional brain networks. The approach shows improved downstream performance across several neuroimaging datasets and phenotypic prediction tasks, such as age, sex, and ASD classification.

**Strengths:**

•	Innovative use of self-distillation and semantic tokenization to learn stable, abstract representations of brain activity.
•	Clear performance improvements over existing foundation models (e.g., BrainLM, Brain-JEPA).
•	Semantic tokenizer proves particularly effective for demographic and clinical predictions (age, sex, ASD).
•	The shift from low-level voxel embeddings to network-based embeddings is conceptually strong.
•	Significant ablation studies to explore the benefit of each component into the model, which shows robust evaluations by the authors.

**Weaknesses:**

•	While results are solid, gains from other architectural components beyond the tokenizer are more modest.
•	The paper could discuss temporal resolution more thoroughly — would finer sampling (e.g., sub-2s TR) lead to better representations or unnecessary noise amplification?
•	The work focuses exclusively on resting-state data; some commentary on potential extension to task-based fMRI would strengthen the contribution.

**Questions:**

1.	Have you tested whether higher temporal resolution (beyond the 2s sampling) affects performance?

---

> ### Author Response · Authors · 2025-11-22
> **Rebuttal Reply 1**
>
> We sincerely thank the reviewer for their evaluation of our manuscript as well as their questions, which has enabled us to significantly improve our work. We were especially pleased that the reviewer considered our work conceptually strong and found the predictive performance improvements convincing.
>
> We briefly summarize the major extensions and clarifications of the revision, before addressing the questions one-by-one.
>
> - **1. Extension to task-based fMRI**: We show strong transfer of learned representations for short fMRI crops of the Hariri emotion task (l.418 and Table 3, p9).
> - **2. Expanded evaluations**: Additional baseline comparison with BrainMass model (Table 2, p8) and extensive ablations on brain atlases (Section A.4.1, p16), temporal resolution (Section A.4.2, p.16), data augmentations (Section A.4.3, p.16), baseline foundation models (Section A.6, p.18).
> - **3. Additional visualisations** of learned network embeddings (Section A.8, p.20), learned kernel properties (Section A.8, p.21), and OOD generalisation (Section A.5, p.17).
>
> Corrections and clarifications:
> - Preprocessing (Filtering): We have updated the manuscript to explicitly state that inputs are bandpass filtered (0.01-0.1Hz). This standard preprocessing step isolates hemodynamic responses and was utilized in our original experiments but not detailed in the initial text.
> - MDBF: We have corrected our description of the MDBF scale as a mood (state) measure rather than personality (trait) measure, and now provide detailed scale descriptions in section A.2.
>
> ---
>
> **Comment 1. While results are solid, gains from other architectural components beyond the tokenizer are more modest.**
>
> We appreciate this observation and would like to clarify the scope of our contributions. Our framework introduces innovations at multiple levels: 1) semantic tokenizer (architectural), 2) global distillation objective across time (training), and 3) teacher-guided curriculum (training). While we indeed use a standard transformer encoder this allows us to leverage highly optimized implementations (e.g., Flash Attention), our ablations (Tables 5 and 7, Figure 4) demonstrate that each component contributes meaningfully to final performance.
>
> Especially the global CLS loss provides substantial gains, demonstrating that the training objective as well as architecture is important. The loss guides the transformer to learn temporally stable, summarizable representations rather than noisy, local patterns. We view the combination of a thorough-studied encoder with specific innovations in tokenization and training to be a strength as it avoids architectural complexity. Nevertheless, we acknowledge that more sophisticated, bespoke encoder designs could potentially further improve results and these indeed represent an interesting direction for future work.
>
> ---
>
> **Comment 2. The paper could discuss temporal resolution more thoroughly — would finer sampling (e.g., sub-2s TR) lead to better representations or unnecessary noise amplification?**
>
> We thank the reviewer for this important consideration. Briefly, we had two primary reasons for using a 2s TR. First, we considered the well-understood sluggishness of the BOLD signal itself. Second, and perhaps more importantly, we recognized that most downstream tasks have TRs close to 2s. As such, if one aims to perform well on resampled, 2s TR data, then we considered we should pretrain on this data too. Nevertheless, we agree this is an important design decision for which we provided no explicit argumentation or empirical evaluation. We are pleased to provide a new ablation study that addresses this question.
>
> For this analysis, we did not resample the UKB data but kept it at its native TR of 0.735s for pretraining. All downstream data was upsampled to this higher resolution instead.
>
> We observe that overall, performance drops mildly as a consequence (average accuracy across 10 tasks) with -1.19%.  Particularly disease classification (ASD -6.45%, MDD -2.5%, SZ -2.42%) performance drops are most severe. While this may relate to the nature of the downstream tasks, it is likely that the relatively high TR of their datasets played a significant role. Indeed, the SRPBS dataset has TRs of 2-2.5s and although ABIDE has a wider range, most are in the 2-3s range. For comparison, HBN has a TR of 0.8s, LEMON of 1.4s, and the UKB of 0.735s. We observe gains for the prediction of cognition (TMT +2.62 and CLVT +0.84) on the LEMON dataset, indicating these phenotypes may benefit from a higher temporal resolution. In sum, we believe that pretraining on resampled data (e.g. with a TR of 2s) is sensible if one aims to evaluate on downstream data which is generally also resampled and has a relatively higher native TR.
>
> We have included this ablation in the manuscript (Section A.4.2, p16).

---

> ### Author Response · Authors · 2025-11-22
> **Rebuttal Reply 2**
>
> **Comment 3. The work focuses exclusively on resting-state data; some commentary on potential extension to task-based fMRI would strengthen the contribution.**
>
> We thank this reviewer for their important feedback. We have now extended our evaluation to task-based fMRI using the Hariri emotion task from UK Biobank, where participants match either shapes or emotional faces in a blocked design. We formulate this as predicting block type ('shapes' vs. 'faces') from short fMRI segments.
>
> This constitutes a challenging transfer despite using UKB data for pretraining. First, task-based fMRI introduces a substantial domain gap: it uses different preprocessing and contains task-evoked activity absent from resting-state data. Second, the task presents a fundamentally different prediction problem, as it requires discriminating within-subject temporal dynamics rather than between-subject phenotypes. Third, task blocks are substantially shorter than pretraining sequences, requiring the model to generate summary representations from limited context.
>
> We address the temporal mismatch by leveraging our masked distillation framework: we construct a single temporal patch from a task block and mask all remaining positions, matching the objective optimized during pretraining. We test three strategies for patch creation (zero-padding, concatenating contiguous blocks, or concatenating timeseries from separate blocks; see Section A.7 (p19) for methodological details).
>
> Table 3 (copied below) shows Brain-Semantoks substantially outperforms Brain-JEPA across all settings, achieving >12% improvement with linear probing} and maintaining strong advantages with finetuning. This demonstrates effective transfer to a different fMRI modality, task paradigm, and temporal scale. We now describe these results in the paper starting at L.417 (p8).
>
>
> | **Model**              | **1 Block**    | **2 Blocks (Cont.)** | **2 Blocks (Cat.)** |
> |------------------------|----------------|----------------------|---------------------|
> | **_Linear Probing_**  |                |                      |                     |
> | Brain-JEPA            | 81.45 ± 0.59  | 82.29 ± 0.28        | 81.06 ± 0.81       |
> | Brain-Semantoks       | **93.84 ± 0.36** | **94.34 ± 0.75**    | **96.50 ± 0.15**   |
> | **_Finetuning_**      |                |                      |                     |
> | Brain-JEPA            | 91.04 ± 1.56  | 92.33 ± 1.86        | 94.71 ± 0.85       |
> | Brain-Semantoks       | **96.89 ± 0.73** | **97.85 ± 0.86**    | **97.70 ± 0.80**   |
>
> ---
>
> We thank the reviewer again for their help in evaluating our paper.

---

### Official Review · Reviewer_mRvP · 2025-11-02

**Soundness:** 3
**Presentation:** 3
**Contribution:** 3
**Rating:** 6
**Confidence:** 4

**Summary:**

The manuscript proposes a self-supervised foundation model for fMRI time series based on a transformer architecture. It introduces a semantic tokenizer that aggregates ROI time series into network-level tokens. The model is trained using a combination of three objectives: a global loss that minimizes the Euclidean distance between global summary tokens from different views, a network token loss that aligns masked and unmasked network tokens, and a temporal regularizer that uses time-averaged representations across each network. Experiments on several neuroimaging datasets (UK Biobank, ABIDE, HBN, SRPBS, and LEMON), show that the proposed model performs better than previous self-supervised and supervised methods.

**Strengths:**

- The model is evaluated on multiple datasets, including UK Biobank, ABIDE, HBN, SRPBS, and LEMON, and shows consistent improvements over strong baselines such as BrainLM and Brain-JEPA.
- The paper also includes extensive ablation studies on the effects of the semantic tokenizer design, temporal regularizer duration, masking type, loss components, and masking ratio.

**Weaknesses:**

- A main weakness is that all experiments rely solely on resting-state fMRI data, which limits the claim of being a true foundation model.
- Tables 1 and 2 need to have a statistical comparison between the best model and other models. It is commonly done with a pairwise test. Then, p-values are usually corrected for multiple comparisons.

**Questions:**

- The paper does not specify how checkpoints were selected, and it is unclear whether the ablation results are based on the validation set, as we can not select hyperparameters on the test set. Please clarify your checkpoint strategy.
- It seems the authors used the wrong ICLR LaTeX template or edited it to fit more content.

---

> ### Author Response · Authors · 2025-11-22
> **Rebuttal Reply 1**
>
> We sincerely thank the reviewer for their evaluation of our manuscript as well as their questions, which has enabled us to significantly improve our work. We were especially pleased that the reviewer appreciated our strong results and extensive set of ablation results.
>
> We briefly summarize the major extensions and clarifications of the revision, before addressing the questions one-by-one.
>
> - **1. Extension to task-based fMRI**: We show strong transfer of learned representations for short fMRI crops of the Hariri emotion task (l.418 and Table 3, p9).
> - **2. Expanded evaluations**: Additional baseline comparison with BrainMass model (Table 2, p8) and extensive ablations on brain atlases (Section A.4.1, p16), temporal resolution (Section A.4.2, p.16), data augmentations (Section A.4.3, p.16), baseline foundation models (Section A.6, p.18).
> - **3. Additional visualisations** of learned network embeddings (Section A.8, p.20), learned kernel properties (Section A.8, p.21), and OOD generalisation (Section A.5, p.17).
>
> Corrections and clarifications:
> - Preprocessing (Filtering): We have updated the manuscript to explicitly state that inputs are bandpass filtered (0.01-0.1Hz). This standard preprocessing step isolates hemodynamic responses and was utilized in our original experiments but not detailed in the initial text.
> - MDBF: We have corrected our description of the MDBF scale as a mood (state) measure rather than personality (trait) measure, and now provide detailed scale descriptions in section A.2.
>
> ---
>
> **Comment 1. A main weakness is that all experiments rely solely on resting-state fMRI data, which limits the claim of being a true foundation model.**
>
> We thank this reviewer for their important feedback. We have now extended our evaluation to task-based fMRI using the Hariri emotion task from UK Biobank, where participants match either shapes or emotional faces in a blocked design. We formulate this as predicting block type ('shapes' vs. 'faces') from short fMRI segments.
>
> This constitutes a challenging transfer despite using UKB data for pretraining. First, task-based fMRI introduces a substantial domain gap: it uses different preprocessing and contains task-evoked activity absent from resting-state data. Second, the task presents a fundamentally different prediction problem, as it requires discriminating within-subject temporal dynamics rather than between-subject phenotypes. Third, task blocks are substantially shorter than pretraining sequences, requiring the model to generate summary representations from limited context.
>
> We address the temporal mismatch by leveraging our masked distillation framework: we construct a single temporal patch from a task block and mask all remaining positions, matching the objective optimized during pretraining. We test three strategies for patch creation (zero-padding, concatenating contiguous blocks, or concatenating timeseries from separate blocks; see Section A.7 (p19) for methodological details).
>
> Table 3 (copied below) shows Brain-Semantoks substantially outperforms Brain-JEPA across all settings, achieving >12% improvement with linear probing} and maintaining strong advantages with finetuning. This demonstrates effective transfer to a different fMRI modality, task paradigm, and temporal scale. We now describe these results in the paper starting at L.417 (p8).
>
>
> | **Model**              | **1 Block**    | **2 Blocks (Cont.)** | **2 Blocks (Cat.)** |
> |------------------------|----------------|----------------------|---------------------|
> | **_Linear Probing_**  |                |                      |                     |
> | Brain-JEPA            | 81.45 ± 0.59  | 82.29 ± 0.28        | 81.06 ± 0.81       |
> | Brain-Semantoks       | **93.84 ± 0.36** | **94.34 ± 0.75**    | **96.50 ± 0.15**   |
> | **_Finetuning_**      |                |                      |                     |
> | Brain-JEPA            | 91.04 ± 1.56  | 92.33 ± 1.86        | 94.71 ± 0.85       |
> | Brain-Semantoks       | **96.89 ± 0.73** | **97.85 ± 0.86**    | **97.70 ± 0.80**   |

---

> ### Author Response · Authors · 2025-11-22
> **Rebuttal Reply 2**
>
> **Comment 2. Tables 1 and 2 need to have a statistical comparison between the best model and other models. It is commonly done with a pairwise test. Then, p-values are usually corrected for multiple comparisons.**
>
> We thank the reviewer for this suggestion. We now perform pairwise statistical comparisons using the Wilcoxon signed-rank test with Holm-Bonferroni correction for multiple comparisons. For each task in both tables, we compare the best model pairwise to other models. Due to space constraints, we include significance markers in the main tables and refer to the appendix for the complete set of p-values (Section A.10, p22). We observe that Brain-Semantoks is significantly better (p < 0.05) than both BrainLM and Brain-JEPA on five tasks for linear probing (Table 1) and seven tasks for finetuning (Table 2). Remaining datasets where Brain-Semantoks scores numerically better but tests are not statistically significant are often due to wider standard deviations resulting from small fMRI datasets.
>
> ---
>
> **Q1. The paper does not specify how checkpoints were selected, and it is unclear whether the ablation results are based on the validation set, as we can not select hyperparameters on the test set. Please clarify your checkpoint strategy.**
>
> We thank the reviewer for raising this important methodological concern. We clarify three aspects of our experimental design:
>
> **Model development data.** Model development (architecture design, augmentation strategies, loss components) was conducted using: a subset of UKB non-overlapping with the holdout set used in our main analyses, a subset of HBN non-overlapping with any samples in this paper, and a large internal (private) dataset. This allowed us to evaluate model transfer with large sample sizes while avoiding data leakage into our test sets. We have added this to Section A.11.
>
> **Checkpoint selection.** Following standard self-supervised learning practice, we pretrain for a fixed number of epochs (100) and always evaluate the final checkpoint. We do not use validation data to select checkpoints, eliminating any risk of overfitting to downstream tasks. This is consistent with recent SSL foundation models which demonstrate that fixed-schedule training obviates the need for validation-based early stopping. We now state explicitly in line 317: "For all analyses we use the model checkpoint following 100 epochs of pretraining.''
>
> **Ablation studies.** The ablations in tables 4-8 are post-hoc analyses on the same test sets used in the main analyses, conducted to provide insight into the relative contributions of different framework components. Importantly, these ablation results did not inform hyperparameter selection. We have clarified this in the main text: "We perform post-hoc ablation studies on core aspects of the Brain-Semantoks framework to analyze the contribution of each component.''
>
> ---
>
> **Q2. It seems the authors used the wrong ICLR LaTeX template or edited it to fit more content.**
>
> We thank the reviewer for bringing this to our attention. Upon investigation, we discovered our preamble inadvertently included the geometry package (copied from a previous project). We did not realize that simply including this package would automatically alter the template's formatting. We have now removed it and verified our revised submission is fully compliant. We apologize for this oversight.
>
>
> ---
>
> Finally, we thank the reviewer again for their help in evaluating our manuscript.

---

### Comment · Area_Chair_8b9R · 2025-12-01

Dear Authors,

It seems that Reviewer sTj7's Q3 and Q4 have not been resolved.
Kindly request a reply to help me better evaluate this work.

Best regards,

AC

---

> ### Author Response · Authors · 2025-12-02
>
> Dear Area Chair,
>
> Thank you. We have indeed addressed Reviewer sTj7's follow-up questions by running the additional control experiments they requested. We just replied to the reviewer detailing the results of these new analyses. We copy the content of that response below for your convenience.
>
> Briefly:
>
> 1. **Q3 (Kernel Control):** We compare our method against the requested "fully learned" control branch. We find that the structured kernel outperforms this control (52.39 vs 51.35), which indicates the benefit of the inductive bias.
>
> 2. **Q4 (No Regularizer):** We empirically confirmed that removing the coding rate regularizer leads to immediate model collapse (cosine similarity of 1.0 between student and teacher), which confirms its necessity.
>
> Best regards,
>
> Authors
>
> &nbsp;&nbsp;&nbsp;
>
>
> **Copy of our reply to Reviewer sTj7**:
>
> We are very pleased to hear that our additional experiments have substantially strengthened our paper and addressed most of the reviewer's initial concerns. We sincerely thank the reviewer for their continued thoughtful evaluation of our work and their control analysis suggestions.
>
> Regarding the control analyses, we agree on both accounts and have run the requested experiments. We summarize the results below:
>
> ---
>
> Q3. **Two-branch kernel control:** To disentangle the effect of the structured design from model capacity, we evaluated the requested "Short + Learned Long-Conv" baseline, using a carefully matched (unstructured) kernel. Specifically, we replaced the structured kernel branch with a standard, learned depthwise convolutional kernel with the exact same receptive field (16) and a comparable parameter count. The results are compared below:
>
> | **Architecture** | **Kernel Configuration**           | **Score** |
> | ---------------- | ---------------------------------- | --------- |
> | **Control**      | Short (3) + Learned Long-Conv (16) | 51.35     |
> | **Ours**         | Short (3) + Structured (16)        | **52.39** |
>
> Despite the greater flexibility of the standard convolution, we observe improved performance with the structured kernel. This confirms that the performance gains are driven by the specific temporal inductive bias (decay profile) rather than simply having a second convolutional path with a larger receptive field. We now include this ablation in table 4 (p10).
>
> ---
>
> Q4. **No-coding-rate variant**: We pretrain without the coding rate regularizer and observed immediate model collapse. Specifically, cosine similarities between student and teacher models of 1 indicate they are identical which prevents learning via self-distillation. This matches the theoretical work in literature such as SimDINO [1] but also prior works such as VICReg [2]. We have included this ablation with training loss visualisations in Appenddix A.13 (p23).
>
>
> [1] Wu, Z., Zhang, J., Pai, D., Wang, X., Singh, C., Yang, J., ... & Ma, Y. (2025). Simplifying dino via coding rate regularization. arXiv preprint arXiv:2502.10385.
>
> [2] Bardes, A., Ponce, J., & LeCun, Y. (2021). Vicreg: Variance-invariance-covariance regularization for self-supervised learning. arXiv preprint arXiv:2105.04906.
>
> ---
>
> Finally, we thank the reviewer again for their work which we believe allowed us to significantly improve our manuscript.

---

### Meta-Review · Area_Chair_8b9R · 2026-01-05

**Summary:**

This paper introduces Brain‑Semantoks, a self‑supervised fMRI foundation model that replaces traditional voxel‑ or ROI‑level reconstruction with a semantic tokenizer to aggregate signals into functional‑network‑level tokens, coupled with a teacher‑guided temporal regularizer (TTR) to stabilize learning on noisy fMRI data. The model is evaluated on multiple resting‑state fMRI datasets and demonstrates consistent improvements over strong baselines in linear‑probe tasks. All reviewers acknowledge the paper’s soundness and clear presentation and the primary concerns raised are limited scope, missing statistical tests, and technical clarifications needed.

**Reviewer Concerns:**

The main concerns raised by the four reviewers have all been resolved, and some minor weaknesses noted can be largely addressable through revisions.

**Reviewer Scores:**

The paper makes a valuable contribution by rethinking tokenization for fMRI through a functional‑network lens and introducing a curriculum‑based temporal regularizer to handle fMRI noise. After rebuttal, reviewers agree that the core idea is innovative and the empirical results are solid. Reviewer C6gR and TTS1 explicitly recommend acceptance, while mRvP and sTj7 indicate that they would not oppose an accept decision.

---

### Decision · Program_Chairs · 2026-01-26

Accept (Poster)